# Cognitive and behavioral effects of whole brain conventional or high dose rate (FLASH) proton irradiation in a neonatal Sprague Dawley rat model

**Michael T. Williams**[1,2,3]*, **Chiho Sugimoto**[2¤a], **Samantha L. Regan**[1,2¤b], **Emily M. Pitzer**[1,2¤c], **Adam L. Fritz**[2], **Mathieu Sertorio**[3,4], **Anthony E. Mascia**[3,4], **Ralph E. Vatner**[3,4], **John P. Perentesis**[1,3,5], **Charles V. Vorhees**[1,2,3]

1 Department of Pediatrics, University of Cincinnati College of Medicine, Cincinnati, OH, United States of America, 2 Division of Neurology, Cincinnati Children's Research Foundation, Cincinnati, OH, United States of America, 3 Cincinnati Children's/University of Cincinnati Proton Therapy and Research Center, Cincinnati, OH, United States of America, 4 Department of Radiation Oncology, University of Cincinnati College of Medicine, Cincinnati, OH, United States of America, 5 Division of Oncology, Cincinnati Children's Research Foundation, Cincinnati, OH, United States of America

¤a Current address: Dept. of Physiology, Michigan State University, East Lansing, MI, United States of America
¤b Current address: Dept. of Human Genetics, University of Michigan Medical Center, Ann Arbor, MI, United States of America
¤c Current address: U.S. Environmental Protection Agency, Research Triangle Park, NC, United States of America
* michael.williams@cchmc.org

**Data Availability Statement:** All relevant data are within the paper and its Supporting Information files.

## Abstract

Recent studies suggest that ultra-high dose rates of proton radiation (>40 Gy/s; FLASH) confer less toxicity to exposed healthy tissue and reduce cognitive decline compared with conventional radiation dose rates (~1 Gy/s), but further preclinical data are required to demonstrate this sparing effect. In this study, postnatal day 11 (P11) rats were treated with whole brain irradiation with protons at a total dose of 0, 5, or 8 Gy, comparing a conventional dose rate of 1 Gy/s vs. a FLASH dose rate of 100 Gy/s. Beginning on P64, rats were tested for locomotor activity, acoustic and tactile startle responses (ASR, TSR) with or without prepulses, novel object recognition (NOR; 4-object version), striatal dependent egocentric learning ([configuration A] Cincinnati water maze (CWM-A)), prefrontal dependent working memory (radial water maze (RWM)), hippocampal dependent spatial learning (Morris water maze (MWM)), amygdala dependent conditioned freezing, and the mirror image CWM [configuration B (CWM-B)]. All groups had deficits in the CWM-A procedure. Weight reductions, decreased center ambulation in the open-field, increased latency on day-1 of RWM, and deficits in CWM-B were observed in all irradiated groups, except the 5 Gy FLASH group. ASR and TSR were reduced in the 8 Gy FLASH group and day-2 latencies in the RWM were increased in the FLASH groups compared with controls. There were no effects on prepulse trials of ASR or TSR, NOR, MWM, or conditioned freezing. The results suggest striatal and prefrontal cortex are sensitive regions at P11 to proton irradiation, with reduced toxicity from FLASH at 5 Gy.

**Funding:** Varian a Siemens Healthineers company, Drs. Michael T. Williams, Charles V. Vorhees, Anthony E. Mascia, Mathieu Sertorio, Ralph E. Vatner, John P. Perentesis. TQL Foundation, Dr. John P. Perentesis. Cure Next Door Foundation, Dr. John P. Perentesis. The funders had no role in study design, data collection and analysis, decision to publish, or preparation of the manuscript.

**Competing interests:** These experiments were funded by Varian, a Siemens Healthineers company that granted the authors intellectual freedom to publish the data. This does not alter our adherence to PLOS ONE policies on sharing data and materials.

## Introduction

Radiation therapy is an important tool for the curative treatment of many brain tumors, but its use is limited by toxicity from exposure of normal brain tissue within the radiation beam path. Neurocognitive impairment is one of the more morbid toxicities of cranial irradiation and is especially pronounced after treatment of pediatric patients [1–4]. Proton radiation can reduce the toxicity of treatment by minimizing exposure of normal brain; however, this benefit is only realized for treating focal targets and confers similar toxicity to X-ray therapy when treatment of the whole brain is required [2, 5–9]. Most advanced proton therapy is delivered by pencil beam scanning. For pencil beam scanning, the treated field is covered by a succession of spots with dose being delivered in a small area at any given time rather than the whole area being irradiated continuously by a scattered field. All the studies on brain FLASH sparring effects have so far used scattered field. Thus it is important to evaluate if spot scanning dose delivery can recapitulate the FLASH effect on brain tissue. Most preclinical studies on the neurocognitive toxicity of proton radiation focused on doses that would be encountered during space flight, much lower than therapeutic doses used clinically [10–13]. More recently, we reported the effects of whole brain irradiation with 11–17 Gy protons at a dose rate of 1 Gy/s on adult rats, that included locomotor ability, rotorod performance, acoustic and tactile startle responses (ASR and TSR, respectively) and a battery of cognitive tests [14]. Treated rats had decreased locomotion for 5 weeks after 17 Gy, but had similar performance in the rotorod, ASR, and TSR compared with sham controls. For cognitive tests, the rats had deficits in egocentric and allocentric learning and memory, but not recognition memory. These data demonstrate that some brain regions may be more sensitive to the effects of proton radiation than others, and therefore multiple domains of learning and memory should be assessed in preclinical models.

New approaches for reducing neurocognitive toxicity are needed for therapeutic regimens requiring whole brain irradiation, such as for the treatment of medulloblastoma, one of the more common brain tumors in children. Recent studies suggest that high dose rates of radiation (>40 Gy/s; FLASH) may confer less toxicity to exposed healthy tissue and may reduce neurocognitive toxicity compared with conventional radiation dose rates (~1 Gy/s). Reduced toxicity in different organ systems from high dose rate radiation (the FLASH effect) has been tested on cell systems, zebrafish, cats, and humans [reviewed in [15]]. However, cognition after FLASH has mainly been tested in mice treated with either electrons or X-rays [16–19]. In mice novel object recognition (NOR) is spared with FLASH dose rates compared with conventional (1 Gy/s) dose rates. These data support the idea that FLASH is less toxic than conventional dose rate radiation. However, the effects of FLASH proton radiation on the brain are not well characterized and have not been modeled for a pediatric population. Furthermore, only incidental or recognition memory has been assessed rather than multiple cognitive domains.

In humans, neuronal differentiation in most regions of the brain is completed prior to birth; in contrast to rats, for which some brain regions continue to develop into the postnatal period [20–22]. Therefore, comparing brain regions between species prior to adulthood is difficult since at a given age some regions in the rat may represent early childhood human development whereas others represent in utero human development. Here we test the hypothesis that FLASH radiation with protons confers less neurotoxicity than conventional dose rate proton radiation in young rats treated with whole brain irradiation on postnatal day (P)11, a period that approximates brain development in early childhood. Females and males were treated with whole brain proton irradiation to a dose of 0, 5, or 8 Gy in a single fraction, comparing a dose rate of 1 Gy/s (conventional) vs. 100 Gy/s (FLASH). The rats were subsequently

assessed in adulthood with a battery of structure/function behavioral tests beginning 53 days after irradiation. At the end of behavioral testing, brain regions of the dopaminergic and glutamatergic systems were assessed.

## Materials and methods

Sprague Dawley CD, IGS rat dams (strain #001, Charles River, Raleigh, NC) with litters of 5 females and 5 males arrived from the supplier on P5 at the Cincinnati Children's Research Foundation's (CCRF's) vivarium that is AAALAC International-accredited and pathogen free. Rats were treated in accordance with protocols approved by the CCRF's Institutional Animal Care and Use Committee (Protocol # IACUC2020-0017) and complied with the ARRIVE guidelines [23]. Dams and litters were initially housed in polysulfone cages (46 cm x 24 cm x 20 cm) in the Alternative Design (Siloam Spring, AR) MACS Flex-air wall mount system with HEPA filtered air supplied at 30 air changes/h and reverse osmosis filtered, UV purified water provided from a Lixit system (SE Lab Group, Napa, CA). Rats were transported to the Cincinnati Proton Therapy Center's vivarium three days prior to proton exposure and housed in PET plastic cages (43 cm x 34 cm x 20 cm) with an Innocage external bottle system (Innovive, San Diego, CA). All cages contained woodchip bedding, a stainless steel hut for enrichment [24], and *ad libitum* access to NIH-07 rat chow (LabDiet #5018, Richmond, IN). Temperature (21 ± 1°C), humidity (50 ± 10%), and light-dark cycle (14:10 h, lights on at 600 h) were automatically regulated in both vivaria. Rats were returned to CCRF vivarium one week after irradiation and housed as described above upon initial arrival. All efforts were made to minimize any pain and suffering.

### Proton exposure

A male and female pair were assigned from each of the 19 litters to one of the 5 groups with the use of a random number table. Rats were individually identified by ear-punch on P7 prior to irradiation. The ProBeam Pencil Beam Scanning Gantry (Varian Medical Systems, Palo Alto, CA, USA) was used to deliver a monoenergetic, single-layer transmission radiation field, such that the target is irradiated with the plateau of a Bragg peak and the Bragg peak itself stops outside of the rat's body. FLASH dose rates were delivered at 250 MeV, while conventional dose rates were delivered at 245 MeV. Details of proton delivery, dosimetry, and monitoring are as described [25]. Rats were irradiated on P11 under 2–4% isoflurane anesthesia with room air using the SomnoSuite system (Kent Scientific, San Diego, CA). Rats were placed in a prone position under the proton beam nozzle to irradiate the whole brain. Field flatness, symmetry and location was quality assured using gafchromic film prior to irradiation. Absolute doses and dose rates were calibrated using a NIST-traceable parallel plate ionization chamber. Dose calculation data using ionization chambers were performed using the dose-to-water formulism as reported in the IAEA TRS 398. The relative biological effectiveness for transmission or plateau irradiations was estimated to be 1.0 for the dose calculations. The absolute dose and dose rate tolerances were 3% and 5%, respectively. Using the spot scanning delivery, the whole brain was uniformly irradiated with a spot pattern that corresponds to a rectangular 2.5 cm x 3.0 cm field. The gantry-mounted laser alignment system was used to localize the brain of each rat using osseous anatomical landmarks, such that the inferior edge of the field and the inferior/posterior edge of the rat brain were aligned. Rats received total doses of 0 (Control), 5, or 8 Gy proton radiation in a single fraction. Two dose rates were used for the 5 and 8 Gy groups, the conventional rate was 1 Gy/s at 245 MeV (Conv) and the FLASH rate was 100 Gy/s at 250 MeV. For the purposes of this study, the dose rates reported are defined as the total dose divided by the total irradiation time, knowing that the instantaneous dose rates are 5–8 times

higher. After exposure, rats recovered from anesthesia prior to being returned to the dam and littermates. Napa Nectar pouches (Systems Engineering Lab Group, Napa, CA) were supplied in the cages as a secondary source of hydration until the rats were ~P56. Body weights were recorded on P7, P11 at the time of exposure, and weekly thereafter until P119. Dams were removed from their litters on P28, since rats are naturally weaned between P25-30 [26, 27], and pair-housed.

## Behavior

Behavioral testing began 53 days after irradiation on P64 and lasted until P132. Non-water maze equipment was cleaned between rats with EPA approved Process NPD solution (Steris, Mentor, OH). Behavioral testing was conducted during the light cycle in the Animal Behavioral Core of CCRF. Personnel performing testing were blind to group membership. Brain tissues were collected at the end of testing. The experimental timeline can be found in Fig 1.

**Open-field.** On P64 rats were tested in PAS activity chambers (41 cm x 41 cm: San Diego Instruments (SDI), San Diego, CA) for 60 min as described [28]. Beam-break data were recorded from an array of 16 x 16 infrared beams in 5 min intervals. Dependent measures were total ambulation and center ambulation. Ambulation is measured as consecutive beam breaks. Rats were returned to their home-cage after testing.

**Acoustic and tactile startle.** ASR and TSR were measured the day after open-field on P65 and P66 in SR-LAB sound-attenuating test chambers (SDI) as described [29]. Rats were placed in SDI large enclosure acrylic cylindrical holders mounted on a platform with a piezoelectric accelerometer attached to the underside. Prior to each session, there was a 5 min adaptation period in the holders. The house light and fan were turned on for the adaptation period and for testing. For ASR the pulse was a 20 ms, 120 dB SPL mixed frequency white noise burst (rise time 1.5 ms). For TSR the pulse was a 20 ms, 60 psi air-puff directed at the dorsum of the rat. The recording window lasted 100 ms from the onset of the pulse. Maximum startle amplitude ($V_{max}$) was measured in mV. At the beginning of each trial any movement present was subtracted from $V_{max}$. Rats were given 5 acoustic (120 dB) trials alternating with 5 tactile (60 psi) trials for a total of 100 trials (50 of each type) on each day.

**Acoustic prepulse inhibition of startle.** On P67 rats were tested for ASR and TSR, using the same conditions as above, except there was an acoustic prepulse prior to the acoustic or

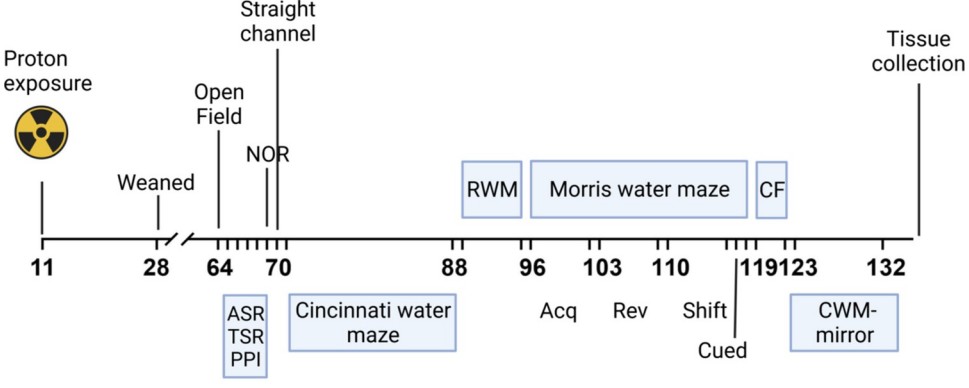

**Fig 1. Experimental timeline.** The experimental timeline in postnatal days. Abbreviations: ASR- acoustic startle response; TSR- tactile startle response; PPI- prepulse inhibition; NOR- novel object recognition; RWM- radial water maze; CF- conditioned freezing; CWM- Cincinnati water maze. Created with BioRender.com.

tactile pulse. The prepulses were 0, 59, 70, 80, or 93 dB SPL mixed frequency white noise burst that preceded the pulse by 70 ms from onset to onset and each prepulse lasted 20 ms. Each rat received a 10 x 10 Latin square sequence of 5 trial types for ASR and TSR and this was repeated twice. $V_{max}$ was the dependent measure.

**Light prepulse inhibition of startle.** On P68 rats were tested for ASR and TSR, using the same conditions as above, with two exceptions: 1) house lights were off and 2) before the acoustic or the tactile pulse there was a light (~1110 lux) prepulse presented at interstimulus intervals of 30, 70, 100, or 400 ms or no light before the pulse as described [30]. The light pre-pulse was from an LED array of high intensity lights (SDI) that provided approximately 1110 ± 17 lux (mean ± SEM) measured at the level of the animal holder with a light meter (Extech Easyview 33, Boston, MA). Light prepulses do not elicit a startle response by them-selves in rats [31]. A 10 x 10 Latin square of each prepulse/pulse type was repeated until 20 tri-als of each were obtained. $V_{max}$ was the dependent measure.

**Novel object recognition.** On P69 NOR was tested in acrylic AnyBox apparatus (40 cm x 40 cm x 40 cm high: Stoelting Co., Wood Dale, IL) with a camera mounted above attached to a computer that ran Any-maze to track movement (Stoelting Co.). For the familiarization phase, rats were placed in the test arena with four identical objects, one in each corner. When rats were within 1 cm of an object, Any-maze scored this as exploration time. To ensure that rats had comparable object exploration, rats had to have 30 s of cumulative exploration of the objects within 10 min [32]. One female 5-Conv rat and one male 8-FLASH rat failed to reach criterion in familiarization and therefore were excluded in the NOR analysis. One hour after familiarization, rats were returned to the arena that now had three identical copies of the origi-nal objects plus one new object. Rats again accumulated 30 s of object exploration up to a trial limit of 10 min. The percent time spent with the novel object (novel object time / 30 s) was the dependent variable. Chance exploration was considered 25% of total exploration time.

**Straight channel.** On P70 rats were tested in a straight channel (244 cm long x 15 cm wide x 50 cm deep) filled half way with water (21 ± 1˚C) with a submerged (~1–2 cm) escape platform at one end. Each of the 4 consecutive trials started with the rat placed facing the wall in the opposite end of the channel from the platform. Trial limit was 2 min. The straight water channel teaches rats that escape is possible, ensures comparable swimming ability prior to learning in the mazes, and is a necessary step for rats to learn in the CWM. Latency to escape was recorded.

**Cincinnati water maze.** From P71-P88 rats were tested in the CWM (configuration A). The CWM consists of 10 T-shaped cul-de-sacs branching from a channel (15 cm wide) extend-ing from the start to the goal [33]. The maze is 51 cm tall and filled halfway with water (21 ± 1˚C). To ensure that rats could only use egocentric learning to navigate the maze, they were run under infrared light. An infrared sensitive camera was mounted above the maze and connected to a monitor in an adjacent room where an experimenter scored latency to escape and errors. Prior to testing, rats were acclimated to the dark for at least 5 min and given 2 tri-als/day with a 5 min limit/trial for 18 days. If a rat did not find the goal, it was removed from the maze after 5 min and not guided to the escape platform. The second trial started after at least a 5 min rest period for rats that did not find the goal on trial-1, but started immediately after trial-1 for rats that found the goal. An error was counted when the head and forelimbs entered into the stem of any dead-end cul-de-sac or into the crossing-arm of a "T". The latency to find the submerged platform and errors were the dependent variables. An error correction for all trials that reached the 5 min limit was the number of errors from the rat with the greatest number of errors.

**Radial water maze.** From P89- P95, working memory was assessed in the RWM as described with modification [34]. A black 210 cm diameter polyethylene tank with 8 arms that

were 55 cm long and 17 cm wide was filled halfway with room temperature water (21 ± 1˚C). Distal cues included posters mounted on the walls of the room. On the first trial of each of the 8 days, all arms had submerged platforms and rats were started in the center of the maze. Rats were given 8 trials/day and allowed up to 2 min to locate a platform. Once a platform was found, the rat was removed from the water and placed in a dry cage for 30 s during which time the platform that was found was removed. Any entry into an arm without a platform was counted as a working memory error. Similar to an error in the CWM, an error was defined as head and forelimbs crossing an imaginary line at the entrance of an arm. Rats that reached the time limit on a trial were given an error correction of the rat with the greatest number of errors in the study. Only days 1–2 were analyzed since rats learned a pattern to escape over days and therefore were not using working memory.

**Morris water maze.** From P96- P118 rats were tested in four phases of the MWM: acquisition, reversal, shift, and cued [35–37]. The maze was a circular pool 244 cm in diameter and 51 cm deep with a conical bottom, filled with room temperature water (21 ± 1˚C) to a depth of 25 cm with a black platform submerged 2 cm below the surface in one quadrant of the pool dependent upon phase. The platform was black and therefore camouflaged against the black background of the pool. The walls around the maze had posters and geometric shapes. For acquisition, reversal, and shift there were 4 platform trials/day for 6 days and 2 probe trials (no platform). The time limit on learning trials was 2 min/trial and the intertrial interval was 5 s spent on the platform plus the amount of time to run other rats in rotation (3–10 min). Each hidden platform phase used smaller platforms to increase the spatial acuity [35, 37]. For acquisition the platform was 10 cm diameter and located in the southwest quadrant (north was defined as the position furthest from the experimenter), for reversal the platform was 7 cm diameter and located in the northeast quadrant, and for shift the platform was 5 cm diameter and located in the northwest quadrant. If rats did not locate the platform in the 2 min period they were removed from the water and placed on the platform. For the learning trials, latency and path efficiency were analyzed. Path efficiency is calculated as a straight line from the start to the platform divided by the path the rat took to reach the platform and is independent of swim speed. One probe trial was given at the beginning of day-3 before the day-3 learning trials and the other on day-7. The time limit for probe trials was 45 s. For probe trials, dependent measures were average distance from the former platform position [38] and number of site crossovers. The cued-random phase had 4 trials/day for 2 days, and a submerged 10 cm platform was marked by a ball on a rod that protruded 10 cm above the water. Distal cues were blocked by black curtains that were drawn around the pool. The platform and start positions were moved on each trial so that spatial cues could not be used to triangulate the location of the platform [35]. The dependent measure for cued-random was latency to the platform since tracking with the curtains closed was not possible. These phases test explicit/spatial/allocentric learning and reference memory (acquisition), cognitive flexibility (reversal), and cognitive flexibility with greater retroactive interference (shift). The cued-random phase is a form of egocentric learning that controls for vision, ability to use proximal cues, motivation to escape, and swimming ability.

**Conditioned freezing.** From P119-P122 rats were run in conditioned freezing using SDI test chambers as described [29]. The chambers were 25 cm x 25 cm with infrared beam arrays and had rat scaled grid floors inserted that were connected to a scrambled shock source. The test chambers were placed inside sound attenuating boxes that had cues mounted on the inside walls. There were four days of testing as follows: Day-1 conditioning, Day-2 contextual response, Day-3 cued freezing with extinction, and Day-4 was cued recall. On Day-1 rats were given 6 min to habituate to the chambers. After the habituation period, rats were given 9 tone-light-foot-shock pairings of 85 dB, 2 kHz, overhead light and shock. Each pairing consisted of

10 s tone-light pairings with a 1.3 mA foot-shock delivered through the grid floor during the last 2 s and 30 s separated each pairing. On Day-2 rats were returned to the same chamber as on Day-1 for 6 min, but without any stimuli present. On Day-3 rats were placed in novel black hexagonal enclosures that fit inside the original arenas with a similar floor space and given a 3 min habituation period. After the habituation, the light-tone stimuli was presented for 30 s for a total of 10 times with a 30 s period of no stimulus between trials. On Day-4 there were 5 light-tone presentations for 30 s with a 30 s period of no stimulus between each light-tone presentation. Beam breaks were recorded for all phases as the dependent variable.

**Cincinnati water maze-mirror.** From P123-P132, the CWM was rerun, however the maze was a mirror image of the first maze (configuration B). All other parameters were the same with errors and latency as the dependent variables.

## Brain tissue collection and analyses

Within 2 weeks of behavioral testing, rats were rapidly decapitated without anesthesia and brains collected. Neostriatum and hippocampus were dissected bilaterally over ice as described [39] using a brain block to obtain 2 mm slices. Each region was placed in conical tubes, rapidly frozen on dry ice, and stored at -80˚C until assayed with ELISA or western blots.

**ELISAs.** Male and female rats from 8 different litters were used for tyrosine hydroxylase (TH), dopamine receptor D1 (DRD1), and dopamine transporter (DAT) ELISAs. Rat ELISA kits were purchased from MyBioSource (San Diego, CA, USA): TH (catalogue number MBS2501697), DRD1 (catalogue number MBS751976), and DAT (catalog number MBS723170). Prior to assay, tissue was weighed and homogenized with RIPA buffer on ice according to the manufacture's specifications. All procedures were carried out in accordance with the manufacture's guidelines. Calibration curves of known standard concentrations were used to determine the unknown concentrations. The results are expressed as ng/mg protein. The limit of detection for all kits was 0.1 ng/mL and no values fell under this limit. Three samples for the TH assay were removed from the analyses since they were over 2 standard deviations from the group mean. This included 1 male and 1 female in the 8-FLASH group and 1 male in the 5-Conv group.

**Western blots.** Western blots were used to analyze DRD2 in the neostriatum and NMDA receptor subunit 1 (NMDA-NR1) in the hippocampus as described [28]. Analyses were performed on six rats from each of the groups; actin was used as reference. Frozen tissues were homogenized in immuno-precipitation assay buffer (25 mM Tris, 150 mM NaCl, 0.5% sodium deoxychlorate, and 1% Triton X-100 adjusted to 7.2 pH with protease inhibitor (Pierce Biotechnology, Rockford, IL)): 200 mL for the hippocampus samples and 75 mL for the striatal samples. Protein was quantified using the BCA Protein Assay Kit (Pierce Biotechnology, Rockford, IL), and samples were diluted to 3 µg/µL. Western blots were performed using LI-COR Odyssey (LI-COR Biosciences, Lincoln, NE) procedures. Briefly, 25 µL of sample was mixed with Laemmli buffer (Sigma, USA) and loaded on a 12% gel (Bio-Rad Laboratories, Hercules, CA) and run at 200 V for 35 min in running buffer (25 mM Tris, 192 mM glycine, 0.1% SDS). The gel was transferred to Immobilon-FL transfer membrane (Millipore, USA) in 1X rapid transfer buffer (AMRESCO, Solon, OH) at 40 V for 1.5 h. Membranes were soaked in Odyssey phosphate buffered saline blocking buffer for 1 h and incubated overnight at 4˚C with primary antibody in blocking buffer with 0.2% Tween 20. Membranes were incubated with secondary antibody in blocking buffer with 0.2% Tween 20 and 0.01% SDS for 1 h at room temperature. Western blot analysis in the neostriatum was conducted with the rabbit anti-DRD2 (Ab85367, AbCam, Cambridge, MA) at 1:500 with Odyssey IRDye 800 secondary antibody at 1:3,000 dilution and in the hippocampus with anti-NMDA-NR1 (ab109182, AbCam, Cambridge, MA) at 1:4000 with Odyssey IRDye 800 secondary antibody at 1:3,000 dilution. Mouse anti-actin (Ab3280, AbCam,

Cambridge, MA) at 1:2000 with Odyssey IRDye 680 at a 1:15,000 secondary antibody was used as a loading control. Relative protein levels were quantified using the LI-COR Odyssey scanner and Image Studio software for fluorescent intensity of each sample normalized to actin.

## Statistical procedures

Data were analyzed using mixed linear ANOVA models (SAS Proc Mixed, SAS Institute 9.4 TS, Cary, NC). Dunnett's tests were used to compare the irradiated groups with controls; for these only p-values are provided. All other main effects and interactions were tested with mixed linear ANOVAs and if significant further analyzed using slice-effect ANOVA in SAS that maintains the overall error term in the analyses and Dunnett's tests when irradiation was a factor in the interaction. Repeated measure factors were fit to either the autoregressive moving average or the autoregressive covariance structure depending upon best fit of the corrected Akaike Information Criterion. Repeated measures were week (body weights), day (ASR, TSR, CWM, RWM, MWM, and CWM-mirror), time (locomotion), or trial (ASR/TSR with acoustic or light prepulse, straight water channel). The estimation method for the covariance parameters was by the restricted maximum likelihood method, with the exception of the CWM data where maximum likelihood estimation was used. Kenward-Roger first order estimated adjusted degrees of freedom, that can be fractional, were calculated for Type III ANOVAs [40]. To control for litter effects and overestimation of sex effects, litter and litter x sex were used as random factors in the analyses. For cognitive tests only deficits were predicted. Significance was set at $p \leq 0.05$. Data are presented as least square mean ± SEM.

## Results

### Mortality and body weights

A male and female pair were assigned from 19 litters to one of 5 groups. The groups were control (isoflurane anesthesia with rats placed under the beam nozzle and not irradiated), 5 Gy at 1 Gy/s conventional proton dose rate (5-Conv), 5 Gy at 100 Gy/s proton dose rate (5-FLASH), 8-Conv, or 8-FLASH. Of all the irradiated rats, there was a single death in the male 5-FLASH group; the cause of death was unknown. No other deaths occurred. One litter had a missing male upon receipt from the supplier, Charles River Laboratories, and therefore there were only 18 males in the control group.

Proton exposure affected body weights. The 8-FLASH ($p < 0.0001$) and 8-Conv ($p < 0.0001$) rats weighed less than the controls (**Fig 2A**). The body weights by age and exposure are shown in **Fig 2B** for females and **Fig 2C** for males. There were interactions of exposure × age, $F_{(60, 2441)} = 4.69$, $p < 0.0001$ and exposure × sex, $F_{(4, 166)} = 2.72$, $p < 0.04$. For the exposure x sex interaction the weight reductions were in both females and males in the 8 Gy groups compared with controls. For the exposure x age interaction, there were no differences between irradiated groups and controls from P7-P35. Beginning at P42 until P119, the 8 Gy groups weighed less than controls, whereas the 5-Conv rats weighed less than controls from P84-91 and from P105-119. The 5-FLASH rats did not differ from controls at any age. All rats, regardless of age, gained weight over weeks $F_{(15, 2310)} = 2034.91$, $p < 0.0001$. Males weighed more than females, $F_{(1, 20.3)} = 464.40$, $p < 0.0001$, and this sex difference interacted with age, $F_{(15, 2311)} = 126.53$, $p < 0.0001$. The interaction of exposure × sex × age was not significant.

### Behavior

A behavioral battery of tests was used to reduce the number of rats used, provide a better characterization of effects for this initial experiment [41], and to be more translatable to human

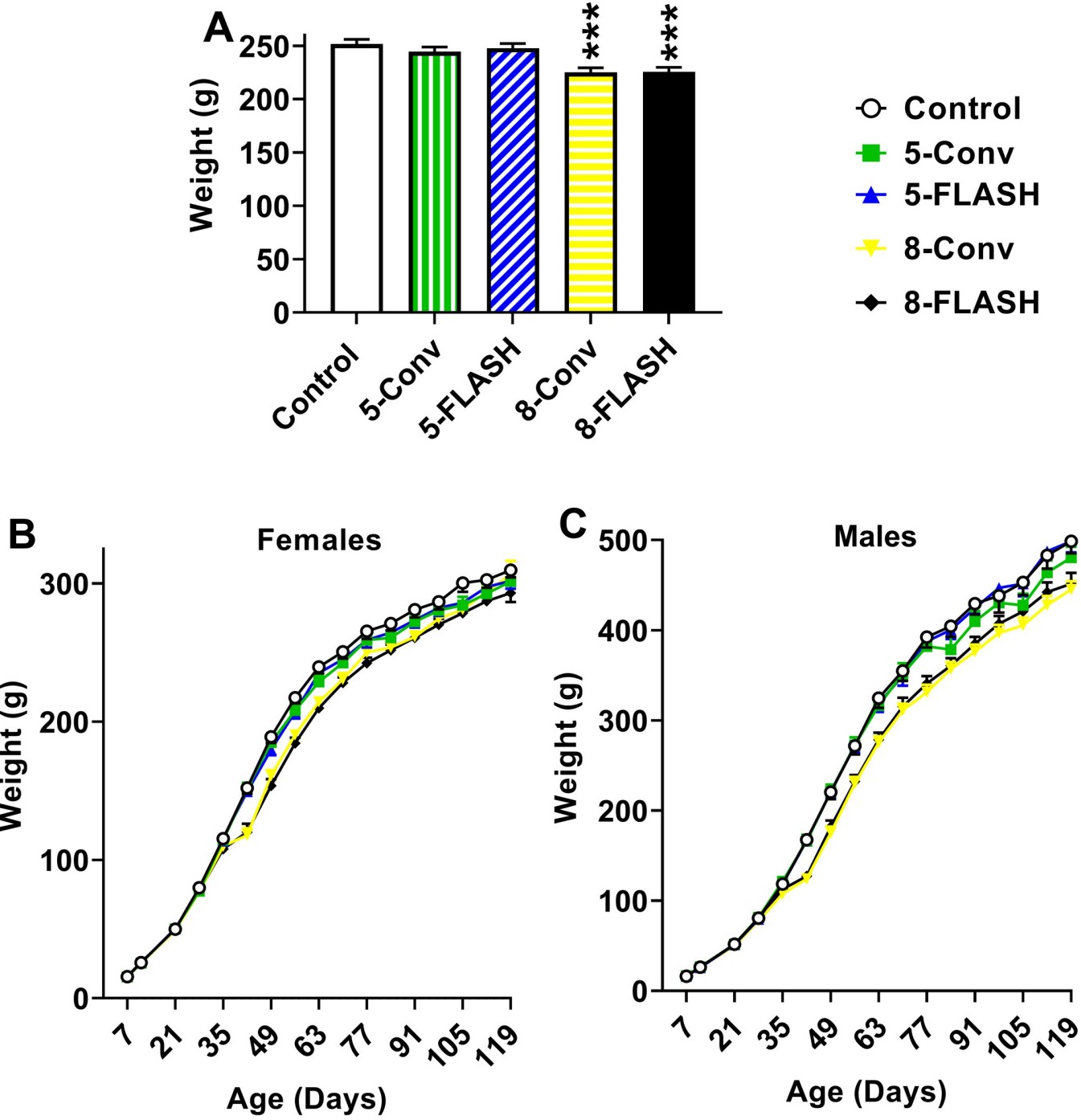

**Fig 2. Body weights [g (mean ± SEM)] of proton irradiated and control rats. A**. The overall effect on body weights by exposure group, averaged across sex and week. **B**. Body weights of females by weeks. **C**. Body weights of males by weeks. While there was an exposure × sex interaction, both females and males dosed with 8 Gy were lighter than controls. See text for a summary of the exposure × age interaction. ***$p < 0.001$ compared with controls. N = 14-19/sex/ exposure at each week. Due to scheduling errors some of the body weights were missed on some weeks.

studies that also employ test batteries of different cognitive domains. While test order may have some effect on subsequent behaviors [42], having a day between different tests alleviated many of these issues [43]. We also minimized these effects by performing tasks from the least

stressful to most stressful (with the exception of the CWM mirror version). While fear potentiated startle can cause changes in dopamine [44], the startle response itself is not known to produce long-term dopamine changes [45]. Behavioral testing began 53 days after irradiation on P64 and lasted until P132 (**Fig 1**). Rats were tested in sequential order as follows: open-field, ASR and TSR, acoustic startle prepulse inhibition (PPI) in combination with ASR or TSR, or light PPI in combination with ASR or TSR, NOR, straight channel swimming, CWM (configuration A), RWM, MWM, conditioned freezing, and CWM-mirror image (configuration B).

**Open field.** Two dependent measures were analyzed for open-field, total ambulation (consecutive beam breaks) and center ambulation. No differences were observed between the irradiated groups and controls for total ambulation (**Fig 3A**). Regardless of group, there was a decrease in total ambulation over the 1 h test period, $F_{(11, 1598)} = 187.59$, $p < 0.0001$ (**Fig 3B**). There was a significant sex × interval interaction, $F_{(11, 1598)} = 2.38$, $p < 0.007$, where females had less ambulation from 6–10 min than males (not shown). No other effects or interactions were significant.

Central ambulation was decreased in the 5-Conv ($p < 0.02$), 8-Conv ($p < 0.02$), and 8-FLASH ($p < 0.02$) groups compared with controls (**Fig 3C**) with no difference for the 5-FLASH group. Females ambulated less than males, $F_{(1, 212)} = 4.82$, $p < 0.03$, and there was a significant interaction of sex × interval, $F_{(11, 1865)} = 3.08$, $p < 0.0005$. Analysis of the interaction showed that females had less ambulation from 6–15 min in the center than males. Ambulation in the center decreased over time regardless of group, $F_{(11, 1865)} = 80.83$, $p < 0.0001$ (**Fig 3D**). No other interactions were significant.

**Acoustic and tactile startle.** For ASR the 8-FLASH group ($p < 0.02$) had smaller $V_{max}$ than controls, but no other group differed from controls (**Fig 4A**). Similarly for TSR, the 8-FLASH group ($p < 0.03$) had smaller $V_{max}$ than controls, whereas no differences were noted for the other irradiated groups and controls (**Fig 4B**). There were no other main effects (sex or day) or interactions for either ASR or TSR.

**Acoustic prepulse inhibition of startle.** For the acoustic PPI, there were no differences between irradiated groups and controls (**Fig 4C**). All groups had a decreased $V_{max}$ with each increase in the acoustic prepulse level, $F_{(4, 554)} = 297.67$, $p < 0.0001$. There were no effects of sex or interactions.

Similarly, for the acoustic PPI with tactile pulses TSR, there were no differences between irradiated groups and controls (**Fig 4D**). There was a decreased $V_{max}$ with each increase in prepulse intensity, $F_{(4, 524)} = 33.37$, $p < 0.0001$. No other main effects or interactions were significant.

**Light prepulse inhibition of startle.** For the light PPI of ASR, there were no differences between irradiated groups and controls (**Fig 4E**). There was a significant difference in $V_{max}$ between each prepulse delay, $F_{(4, 251)} = 261.43$, $p < 0.0001$, and there was a sex × prepulse interval interaction regardless of exposure, $F_{(4, 251)} = 2.71$, $p < 0.04$. Examination of the interaction did not reveal differences between the sexes at any prepulse delay. No other main effects or interactions were significant.

For the light PPI of TSR, there were no differences between irradiated groups and controls (**Fig 4F**). With the exception of the 30 ms prepulse compared with the 100 ms prepulse delays, there was a significant difference between the $V_{max}$ of each prepulse delay, $F_{(4, 385)} = 199.93$, $p < 0.0001$. The sex × prepulse delay interaction was significant, $F_{(4, 385)} = 3.05$, $p < 0.02$ and was the result of males having a greater $V_{max}$ than females with no prepulse but not when prepulses were present. No other main effects or interactions were significant.

**Novel object recognition.** There was no difference between irradiated groups and controls for percent novel object preference and no sex difference or interaction of exposure × sex. The preference mean ± SEM for each exposure group are as follows: Controls, 36.4 ± 3.0%; 5-Conv, 34.6 ± 3.1%; 5-FLASH, 38.8 ± 3.0%; 8-Conv, 43.2 ± 3.0%; 8-FLASH, 35.6 ± 3.1%.

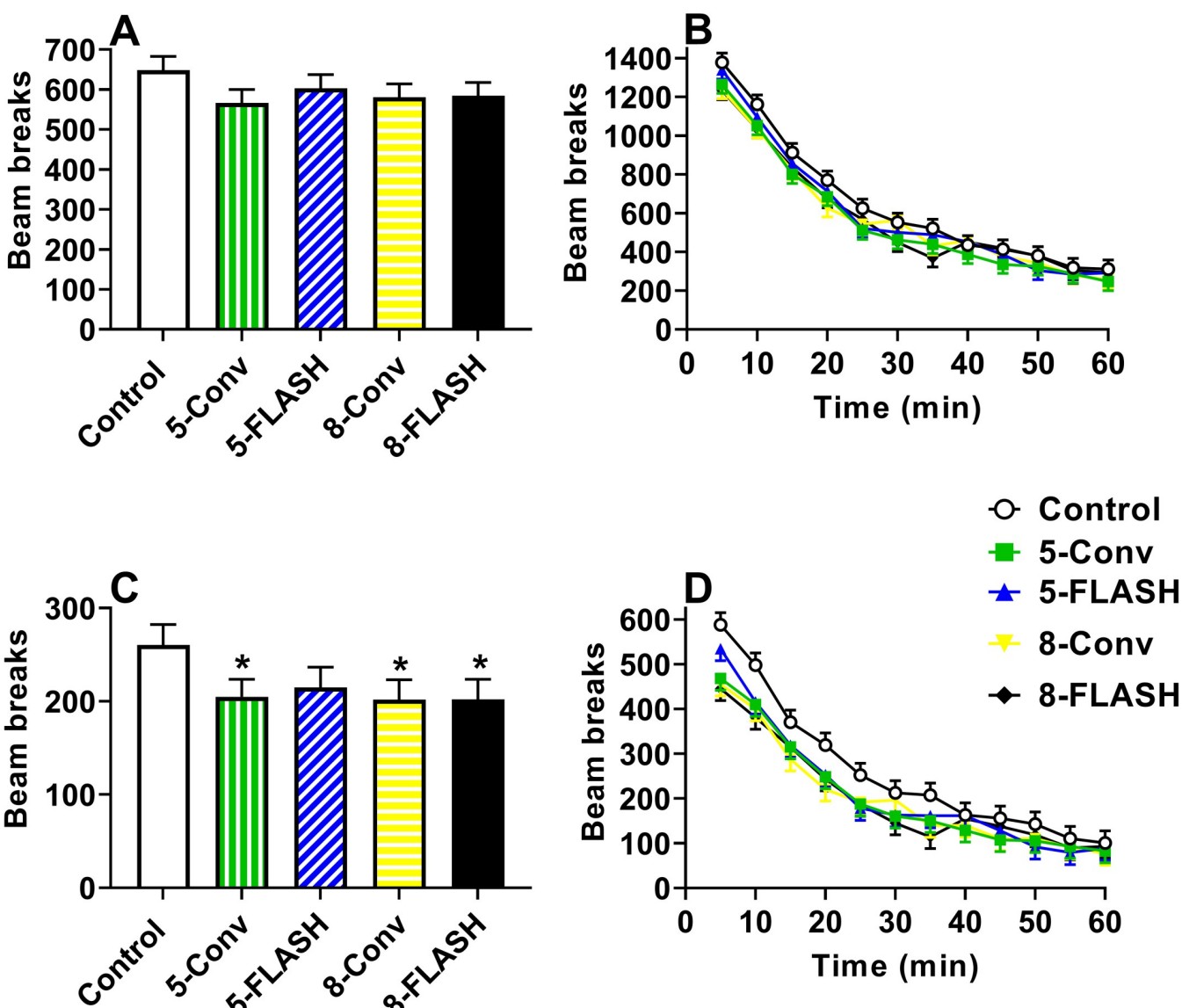

**Fig 3. Locomotor activity [beam breaks (mean ± SEM)] of proton irradiated and control rats. A**. Effects on ambulation by exposure, averaged across sex and test interval (time). No differences in irradiated groups compared with controls. **B.** Ambulation by test interval with males and females combined. **C.** Effects on center ambulation by exposure, averaged across sex and time. All irradiated groups with the exception of the 5-FLASH group had reduced center ambulation compared with controls. **D.** Center ambulation by test interval with males and females combined. *p < 0.05 compared with controls. N = 19/sex/exposure except control male = 17 and 5-FLASH male and female = 18.

**Straight channel.** There was no difference between the irradiated groups and the controls for straight channel swim latency. All groups had a reduction in latency over trials, $F(3, 421) = 38.04$, $p < 0.0001$. There was no difference between sexes and no significant interactions. The latency mean ± SEM for each exposure group over all trials are as follows: Controls, 11.8 ± 0.7 s; 5-Conv, 10.8 ± 0.7 s; 5-FLASH, 11.7 ± 0.7 s; 8-Conv, 10.3 ± 0.7 s; 8-FLASH, 11.3 ± 0.7 s.

**Cincinnati water maze.** All irradiated groups were slower in locating the platform on all days compared with the controls (**Fig 5A**): 5-Conv ($p < 0.003$), 5-FLASH ($p < 0.04$), 8-Conv ($p < 0.0003$), and 8-FLASH ($p < 0.001$). There was an interaction of exposure × day, $F(68, 3096) = 1.47$, $p < 0.008$ (**Fig 5B**). For the first 5 days there were no differences. On Day-6, all

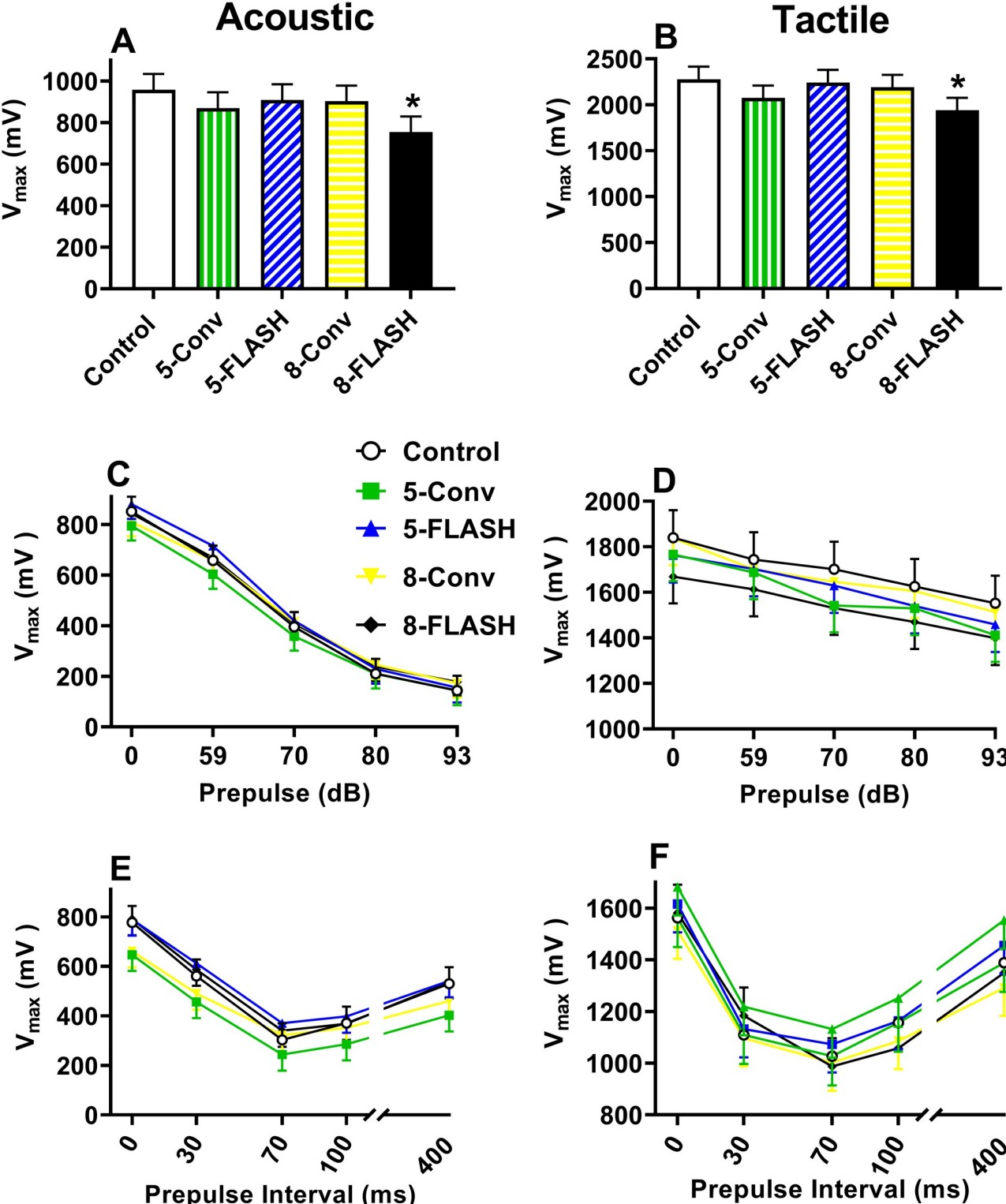

**Fig 4. Acoustic or tactile startle responses [V$_{max}$ (mean ± SEM)] with or without acoustic or light prepulses. A**. The effect on acoustic startle response by exposure, averaged across sex and days. The 8-FLASH group had reduced acoustic startle compared with controls. **B**. The effect on tactile startle response by exposure, averaged across sex and days. The 8-FLASH group had reduced tactile startle compared with controls. For panels A & B, N = 19/sex/exposure except Control and 5-FLASH males = 18/group. **C**. The effect of acoustic prepulses prior to an acoustic pulse by exposure averaged across sex. There were no differences between controls and irradiated groups. **D.** The effect of acoustic prepulses prior to a tactile pulse by exposure

averaged across sex. There were no differences between controls and irradiated groups. For panels **C & D**, N = 19/sex/exposure except 5-FLASH females and males = 18 and Control males = 17. Two rats were found outside of the holder and their data were not used. **E.** The effect of light prepulses prior to an acoustic pulse by exposure averaged across sex. There were no differences between controls and irradiated groups. **F.** The effect of light prepulses prior to a tactile pulse by exposure averaged across sex. There were no differences between controls and irradiated groups. For panels **E & F**, N = 16/sex/exposure except 5-FLASH males = 15 and Control males = 14. The discrepancy in numbers was the result of the computer program failing to save data from rats that were run in the test. $^*$p < 0.05.

the irradiated groups were slower than controls, with the exception of the 5-Conv group, whereas on Days 7–9, all irradiated groups were slower than controls. With the exception of the 5-FLASH group, the other irradiated groups were slower at locating the platform from Days 10–18 than the controls. Regardless of group, all groups showed learning and had a reduction in latency over days, F(17, 2876) = 35.93, p < 0.0001. There were no differences between sex and no other significant interactions.

All the irradiated groups made more errors over all days compared with the controls (**Fig 5C**): 5-Conv (p < 0.003), 5-FLASH (p < 0.04), 8-Conv (p < 0.0002), and 8-FLASH (p < 0.0004). The interaction of exposure × day was significant, F(68, 3098) = 1.46, p < 0.009 (**Fig 5D**). There were no differences from Days 1–4, on Day-5 all irradiated groups made more errors than controls except the 5-Conv, and from Day 6–8 when all irradiated groups made more errors than controls. On Days 9–16 and day 18 all irradiated groups with the exception of the 5-FLASH group made more errors than controls, whereas on Day-17 all groups made more errors than controls. The number of errors decreased over days, F(17, 2880) = 33.84, p < 0.0001, reflecting learning. No difference was noted between sexes or any other interactions.

**Radial water maze.** On Day-1 the 5-Conv (p < 0.0001), 8-Conv (p < 0.02), and 8-FLASH (p < 0.03) groups were slower in locating the platforms than controls, while there was no difference between the 5-FLASH group compared with controls (**Fig 6A**). Regardless of group, there was an increase in latency to locate the platform over trials, F(7, 1071) = 39.23, p < 0.0001 (not shown). For errors per trial on Day-1, compared with controls the 5-Conv (p < 0.002) and the 8-Conv (p < 0.03) groups made more errors and there were no differences in the FLASH groups (**Fig 6B**). Rats committed more errors over trials, F(7, 887) = 63.76, p < 0.0001 (not shown). No effects of sex or other interactions were found for latency or errors.

For Day-2 latency, the 5-FLASH (p < 0.04) and 8-FLASH (p < 0.03) groups were slower at locating the platform than controls, while there were no differences between controls and the other groups (**Fig 6C**). There was an increase in latency to locate the platform over trials, F(7, 1012) = 58.22, p < 0.0001, regardless of group (not shown). On day-2, there were no differences in errors per trial between the irradiated groups and controls (**Fig 6D**). There was a significant interaction of exposure group × sex × trial, F(28, 1154) = 1.54, p < 0.05. For females there were no differences between irradiated groups and controls on any trials. For males the 5-FLASH (p < 0.0007) and 8-Conv (p < 0.0004) groups had more errors on trial-7 than controls. The means ± SEM for the number of errors on trial-7 for males for each exposure group are as follows: Controls, 2.2 ± 0.7; 5-Conv, 3.3 ± 0.7; 5-FLASH, 5.2 ± 0.7; 8-Conv, 5.4 ± 0.7; 8-FLASH, 3.7 ± 0.7. No other trial showed significant group differences. There was an increase in errors to locate the platform over trials, F(7, 1041) = 73.92, p < 0.0001, regardless of group (not shown).

**Morris water maze- acquisition.** For latency there were no differences between the irradiated groups and controls, however there was an interaction of exposure × day, F(20, 832) = 1.56, p < 0.05. There were no differences on Day 1 or on Days 4–6. The 5-Conv and 8-Conv groups took longer to locate the platform on days 2 and 3 compared with controls (**Fig 7A**).

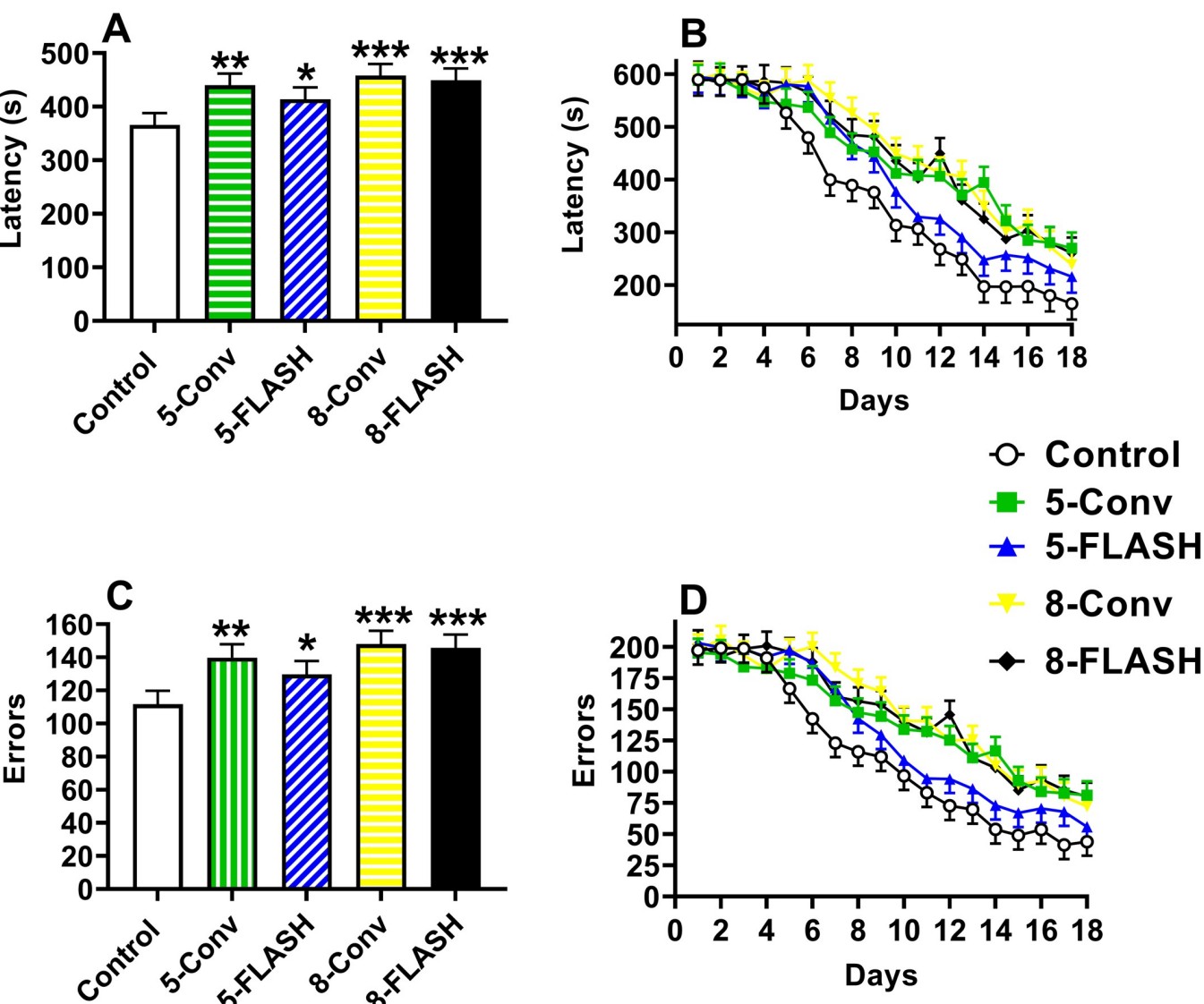

**Fig 5. The Cincinnati water maze latency [s (mean ± SEM)] and number of errors (mean ± SEM) of proton irradiated and control rats. A.** The overall effect on latency where irradiated groups took longer to locate the platform compared with controls. **B.** The learning curves for latency. There was an interaction of exposure x day that is described in the results. Differences between the controls and irradiated groups began after day 5 and from day 10 to 18 all groups except the 5-FLASH group took longer to locate the platform. **C.** The overall effect on errors where irradiated groups had more errors to locate the platform. **D.** The learning curves for errors. Similar to latency there was an exposure x day interaction that is described in the results. N = 19/sex/exposure except control male and 5-FLASH male = 18. *p < 0.05, **p < 0.01, and ***p < 0.001 compared with controls.

All rats had decreased latencies over days, F(5, 719) = 167.56, p < 0.0001. Males found the platform faster than females, F(1, 18.9) = 5.57, p < 0.03 (not shown). No other significant effects were found for latency.

For path efficiency, there were no differences between the irradiated groups and controls (**Fig 7B**). All rats became more efficient at locating the platform over days F(5, 711) = 112.75, p < 0.0001. Males were more efficient than females, F(1, 18.3) = 27.06, p < 0.0001, however this difference was limited to days 3–6, F(5, 711) = 4.54, p < 0.0004 (not shown). No other interactions were significant.

For average distance to the former platform site on probe trials, there were no differences in irradiated groups compared with control group (**Fig 7C**). All groups had smaller average

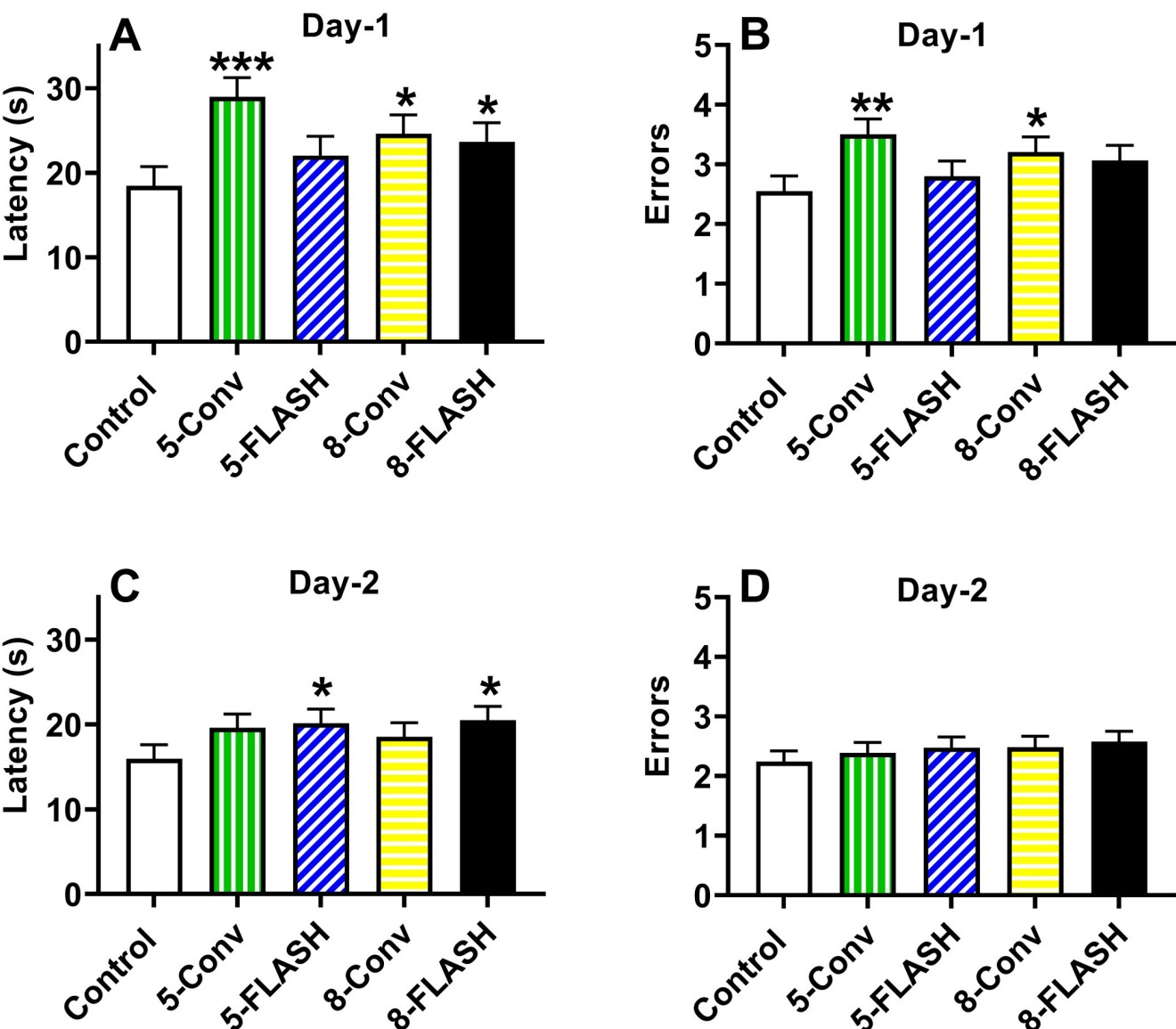

**Fig 6. The radial water maze latency [s (mean ± SEM)] and number of errors (mean ± SEM) of proton irradiated and control rats on Day 1 and Day 2.**
**A.** The overall effect on latency in the radial water maze on Day 1. With the exception of the 5-FLASH group, the irradiated groups took longer to locate the platform. **B.** The overall effect on mean errors per trial in the radial water maze on Day 1. The 5-Conv and 8-Conv group made more errors compared with controls, while there was no difference for the FLASH groups and controls. N = 19/sex/exposure except control male and 5-FLASH male = 18. **C.** The overall effect on latency in the radial water maze on Day 2. The 5-FLASH and 8-FLASH groups took longer to locate the platform compared with controls. **D**. The overall effect on mean errors in the radial water maze on Day 2. There were no differences between irradiated groups and controls. There was an interaction of exposure × sex × trial where males in the 5-FLASH and 8-Conv groups made more errors on trial-7 only (not shown). N = 19/sex/exposure except control male, 5-FLASH male, and 8-Conv females = 18. One 8-Conv female was inadvertently missed. $^{*}p < 0.05$, $^{**}p < 0.01$, and $^{***}p < 0.001$ compared with controls.

distances on day-7 compared with day-3 probe trials, $F(1, 175) = 61.71$, $p < 0.0001$. Males had shorter average distances than females, $F(1, 157) = 37.27$, $p < 0.0001$ (not shown). No interactions were significant.

Similarly for crossovers during the probe trials, there were no differences between irradiated groups and controls (not shown). All groups had more crossovers on day-7 than day-3, $F(1, 178) = 12.84$, $p < 0.0005$, and males had more crossovers than females, $F(1, 35.5) = 4.94$, $p < 0.04$. No interactions were significant.

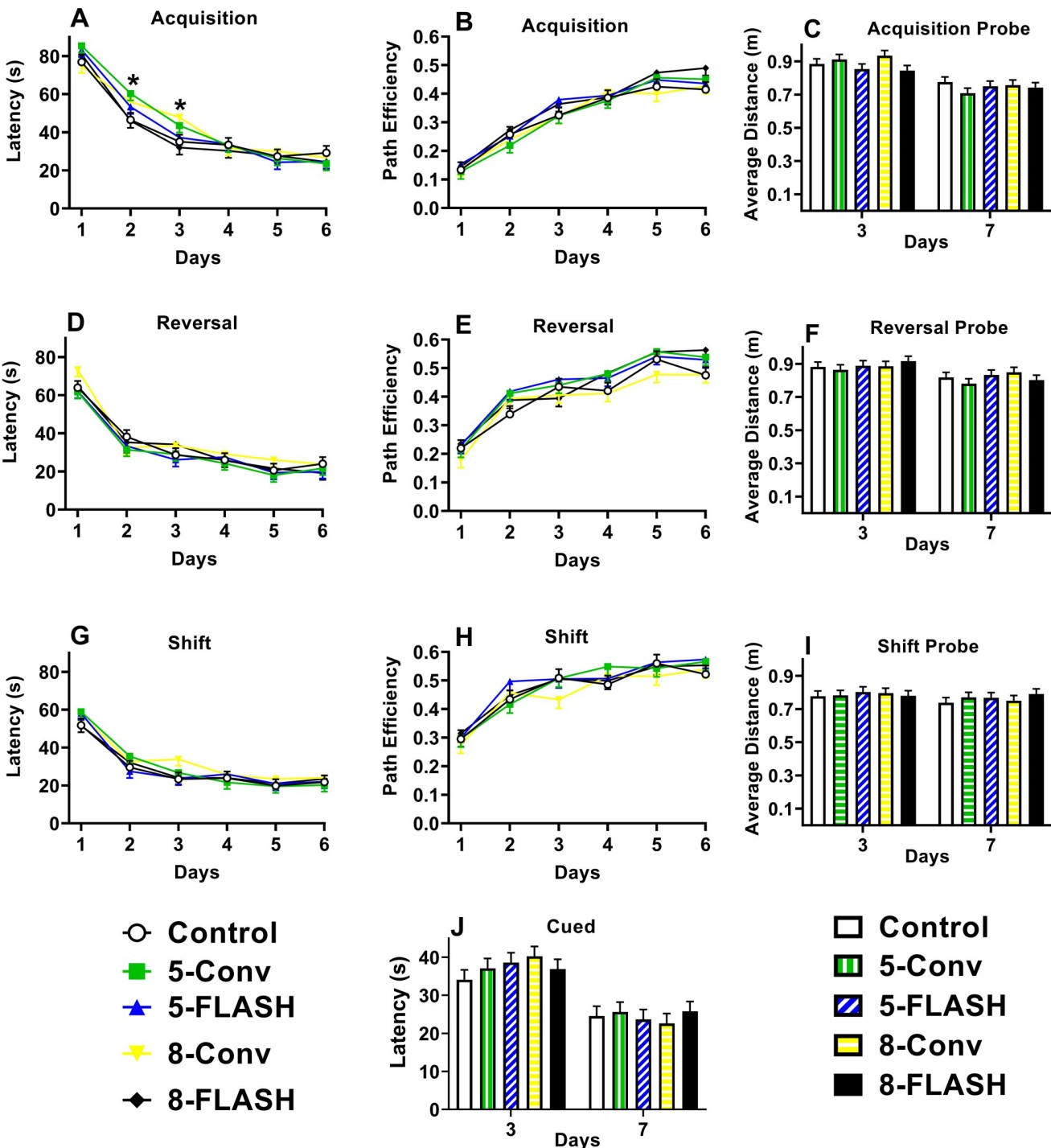

**Fig 7.** The Morris water maze latency [s (mean ± SEM)], path efficiency (mean ± SEM), and average distance from the platform on probe trials [m (mean ± SEM)] for the acquisition (A, B, C), reversal (D, E, F), and shift (G, H, I) phases and latency [s (mean ± SEM)] during the cued (J) phase of proton irradiated and control rats. There were no overall differences between the irradiated groups and controls on any measure in the Morris water maze. There was a significant interaction of exposure × day for acquisition latency where the 5-Conv and 8-Conv groups took longer to locate the platform on days 2 and 3 compared with the controls, *p < 0.05. No other interactions involving exposure were found. N = 19/sex/exposure except control male and 5-FLASH male = 18.

**Morris water maze- reversal.**   For latency (**Fig 7D**) and path efficiency (**Fig 7E**), there were no effects of irradiation exposure compared with controls. Latency decreased, $F(5, 633) = 128.84$, $p < 0.0001$, and path efficiency increased, $F(5, 668) = 92.65$, $p < 0.0001$, over days regardless of group. Males located the platform faster than females, $F(1, 158) = 20.98$, $p < 0.0001$, and were more efficient, $F(1, 18.6) = 57.20$, $p < 0.0001$ (not shown). There were no significant interactions.

For average distance to the platform (**Fig 7F**) and crossovers (not shown) during probe trials, there were no differences between irradiated groups and controls. Average distance declined over days, $F(1, 178) = 21.79$, $p < 0.0001$, and crossovers increased, $F(1, 178) = 8.38$, $p < 0.005$. For average distance, males were closer to the former platform location than females, $F(1,17.4) = 74.88$, $p < 0.0001$, and although there was an interaction of sex × day, $F(1, 178) = 5.64$, $p < 0.02$, males still were closer to the platform regardless of day. For crossovers, males had more crossovers than females, $F(1, 161) = 5.09$, $p < 0.03$.

**Morris water maze- shift.**   For latency (**Fig 7G**) and path efficiency (**Fig 7H**), there were no effects of irradiation compared with controls. Regardless of group, there was a decrease in latency over days, $F(5, 696) = 87.14$, $p < 0.0001$, and there was an increase in path efficiency, $F(5, 681) = 66.23$, $p < 0.0001$. Males located the platform faster, $F(1, 159) = 17.80$, $p < 0.0001$, and were more efficient, $F(1, 18.2) = 69.74$, $p < 0.0001$, than females (not shown). Even though sex × day was significant for path efficiency, $F(5, 681) = 4.27$, $p < 0.0009$, males were more efficient on all days. No other interactions were significant for latency or path efficiency.

For average distance from the platform during the probe trials, there were no differences between irradiated groups and controls (**Fig 7I**). Males were closer to the platform location than females, $F(1, 17.6) = 53.47$, $p < 0.0001$, and although there was a sex × day interaction, $F(1, 177) = 9.74$, $p < 0.003$, the effect was the same. There were no other significant effects for average distance. For crossovers, the 5-FLASH ($p < 0.04$) and the 8-FLASH ($p < 0.008$) groups had fewer crossovers than controls. The means ± SEM for the number of crossovers for each exposure group are as follows: Controls, 1.59 ± 0.14; 5-Conv, 1.32 ± 0.14; 5-FLASH, 1.28 ± 0.14; 8-Conv, 1.32 ± 0.14; 8-FLASH, 1.18 ± 0.14. Regardless of group, rats had more crossovers on day-7 than day-3, $F(1, 178) = 7.16$, $p < 0.009$, and males had more crossovers than females, $F(1, 18.1) = 17.42$, $p < 0.0007$ (not shown). No other effects were significant for crossovers.

**Morris water maze–cued.**   There were no differences in cued water maze performance between the irradiated groups and controls (**Fig 7J**). The latency to reach the platform decreased over day, $F(1, 178) = 96.83$, $p < 0.0001$, regardless of group. There were no significant sex differences or interactions.

**Conditioned freezing.**   For conditioned freezing there were no effects of irradiation on any of the four days of testing. On Day-1 all the rats had reductions in movement between habituation and conditioning, $F(1, 170) = 348.80$, $p < 0.0001$ (**Fig 8A**). There were no effects of exposure, sex, or interactions for Day 1 or for Day 2 the test of contextual memory (**Fig 8B**), or Day 3 habituation (not shown). On Day-3 when the tone was presented to test cued memory, there was a significant effect of interval, $F(9, 1120) = 10.25$, $p < 0.0001$, but not of group with initial decreases in movement and then increased movement at the end of the repeated intervals (**Fig 8C**). There were no significant effect of sex or interactions with the tone on. Similarly for Day-4 with the tone on, there was a slight decrease in movement until the last interval, $F(4, 491) = 5.00$, $p < 0.0007$ (**Fig 8D**). No significant sex effects or interactions were found on Day-4.

**Cincinnati water maze–mirror.**   For latency to the platform, the 5-Conv ($p < 0.008$), 8-Conv ($p < 0.0002$), and 8-FLASH ($p < 0.006$) groups were slower at locating the platform than the controls, but there was no difference between the 5-FLASH group and controls (**Fig 9A**). The latency to locate the platform decreased over days, $F(9, 1397) = 50.98$, $p < 0.0001$ (**Fig 9B**). There was no significant sex main effect, but there was a sex × day interaction, $F(9,$

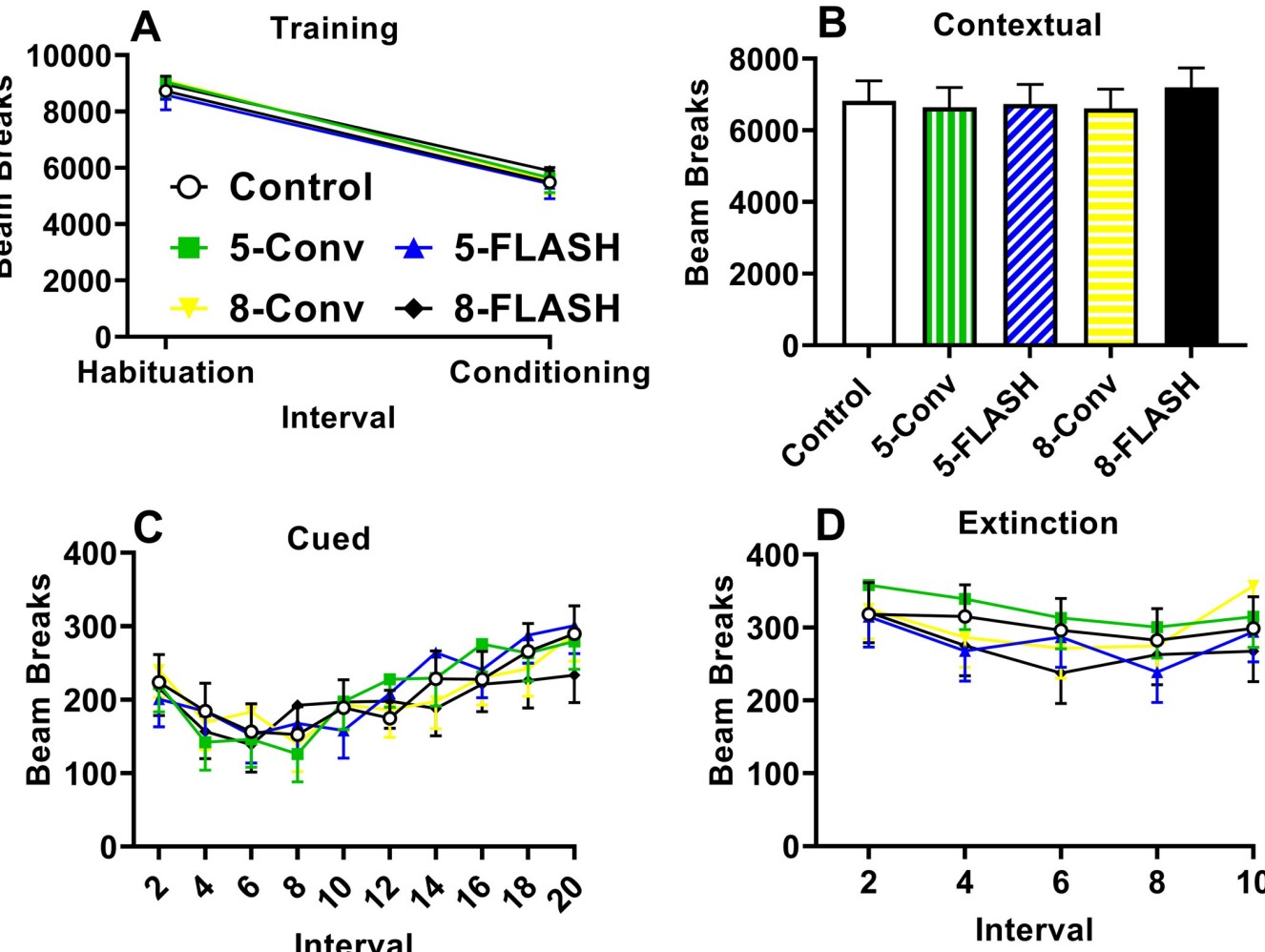

**Fig 8. Conditioned freezing [beam interruptions (mean ± SEM)] for proton irradiated and control rats. A.** Day 1 habituation phase and conditioning phase; there were no differences between the irradiated groups and controls. **B.** Day 2 contextual learning phase; there were no differences between irradiated groups and controls. **C.** Day 3 extinction phase for the tone on trials; no differences between groups. **D.** Day 4 reinstatement phase; no differences were noted. N = 16–19/exposure/sex. Several rats escaped the test arena during the conditioning phase and their data were not used.

1397) = 2.14, p < 0.03. Females located the platform faster on days 1 and 2 than males. No other interactions were significant.

Similar to latency, for errors the number of errors were greater in the 5-Conv (p < 0.02), 8-Conv (p< 0.0001), and 8-FLASH (p < 0.008) groups compared with controls with no difference between the 5-FLASH group and controls (**Fig 9C**). The number of errors decreased over days for all groups, F(9, 1373) = 47.20, p < 0.0001 (**Fig 9D**). No sex differences or other interactions were significant.

## Neurochemical markers

A number of studies examining the effect of FLASH in the brain of adult mice have focused on markers of overt neurotoxicity such as glial fibrillary acidic protein staining or bromodeoxyuridine uptake [16–18]. However, in this study the approach was to examine the neurotransmitter system related to the functional changes observed in these rats. Therefore the focus was on the dopamine system because of the changes in the CWM and the NMDA receptor subunit 1

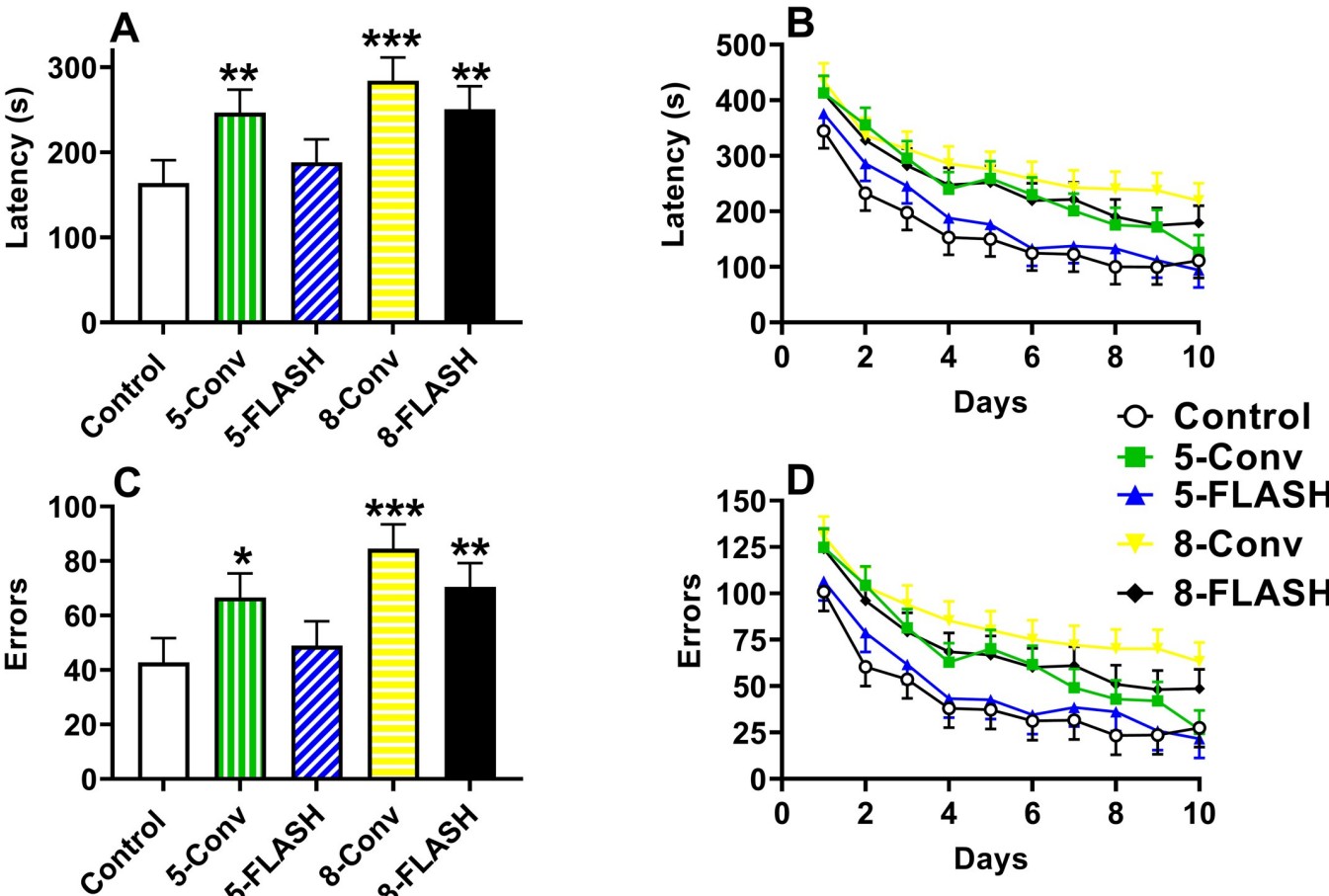

**Fig 9. The Cincinnati water maze mirror version latency [s (mean ± SEM)] and number of errors (mean ± SEM) of proton irradiated and control rats. A.** The overall effect on latency where irradiated groups with the exception of the 5-FLASH group took longer to locate the platform compared with controls. **B.** The learning curves for latency. All exposure groups had decreased latencies over days. **C.** The overall effect on errors where irradiated groups with the exception of the 5-FLASH group had more errors to locate the platform compared with controls. **D.** The learning curves for errors. Errors decreased over days for all groups. N = 19/sex/exposure except control male and 5-FLASH male = 18. $^*p < 0.05$, $^{**}p < 0.01$, and $^{***}p < 0.001$ compared with controls.

(NMDA-NR1) in the hippocampus to confirm the MWM results. Tyrosine hydroxylase (TH), dopamine transporter (DAT), and dopamine receptor D1 (DRD1) in the neostriatum were assessed by ELISA. For TH, there were no differences between irradiated groups and controls nor a difference between sexes. The exposure × sex interaction was significant, $F(4, 48.7) = 3.99$, $p < 0.007$. For females, the 5-FLASH ($p < 0.02$) and 8-FLASH ($p < 0.02$) groups had increased levels of TH compared with control females (**Fig 10A**). There were no differences in the males. For DAT the 5-Conv group ($p < 0.0003$) had higher levels compared with controls while no other group differed from controls (**Fig 10B**). Males had greater DAT levels than females, $F(1, 63) = 15.81$, $p < 0.0002$ (not shown). For DRD1, the 8-Conv ($p < 0.04$) and 8-FLASH ($p < 0.01$) groups had increased levels compared with controls with no difference between controls and the 5-Conv and 5-FLASH groups (**Fig 10C**). Males had greater levels of DRD1 than females, $F(1, 7) = 17.11$, $p < 0.005$ (not shown). The exposure × sex interaction was significant as well, $F(4, 56) = 2.55$, $p < 0.05$ (**Fig 10D**). For the females, the 8-FLASH group had greater levels of DRD1 than female controls. There was no difference in males.

Western blots were done for DRD2 in the neostriatum and NMDA-NR1 in the hippocampus. For DRD2 in the neostriatum, there were no differences between the irradiated groups

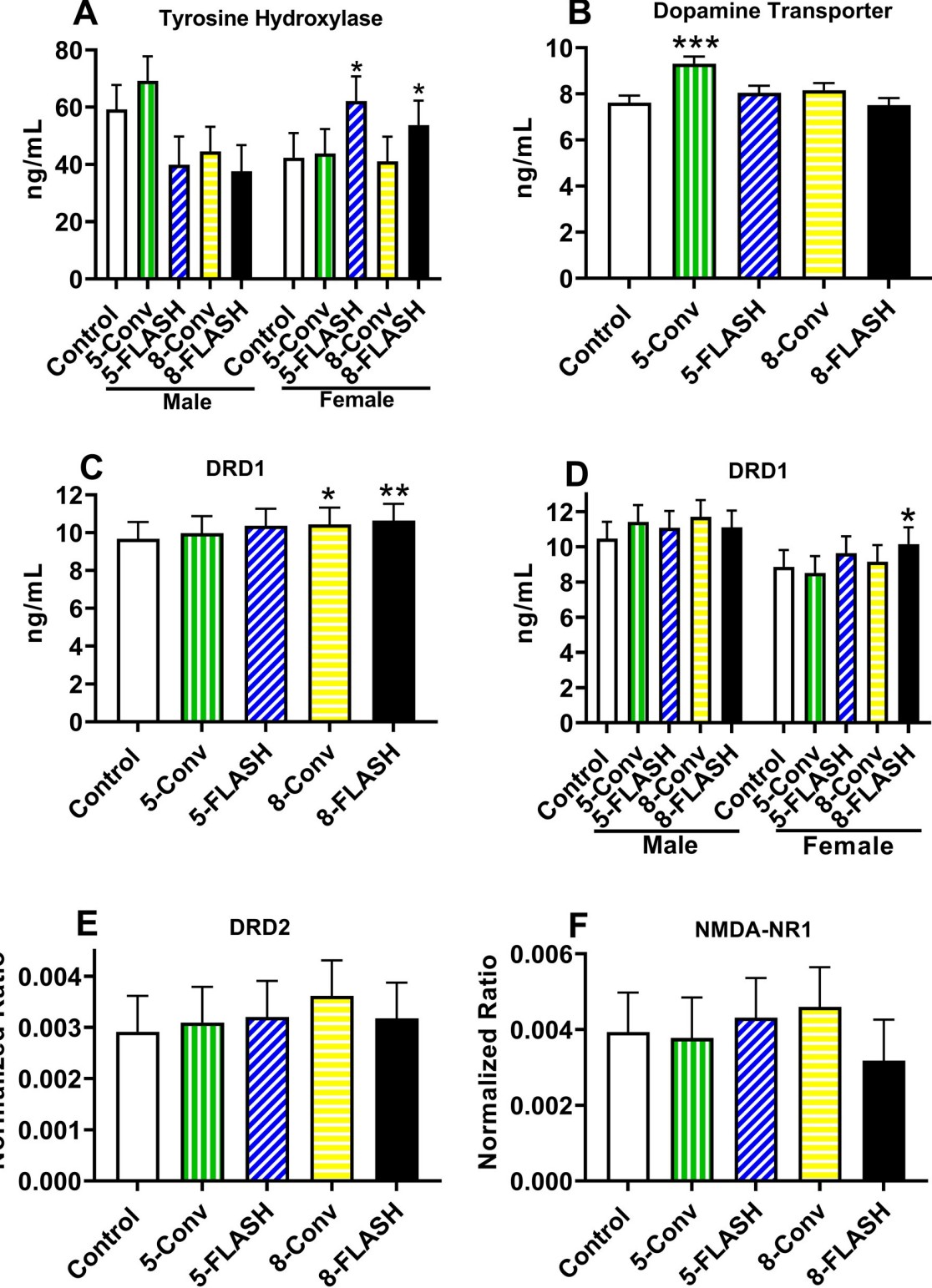

**Fig 10.** Physiological measures in irradiated and control rats in the neostriatum (A-E) and hippocampus (F). **A.** For TH there was no overall difference between irradiated groups and controls, but there was an exposure × sex interaction. There were no differences in males whereas the FLASH females, regardless of dose, had increased TH levels compared with control females. **B.** For DAT the 5-Conv group had increased levels of DAT compared with the controls. **C.** For DRD1 there was an increase in receptor levels for the groups that received 8 Gy, regardless of dose rate, compared with controls. **D.** There was an exposure × sex interaction. There were no

differences in males, whereas the 8-FLASH females had increased DRD1 levels compared with control females. ELISA was used for Panels **A-D**, N = 8/sex/exposure. **E.** There were no differences in DRD2 in the neostriatum. **F.** There were no differences for NMDA-R1 in the hippocampus. Westerns were used for panels **E-F**, N = 6-7/sex/exposure.

and controls (**Fig 10E**). There were no sex differences or an interaction of irradiation × sex. Similarly, for NMDA-NR1 in the hippocampus, there were no differences between the irradiation groups and controls (**Fig 10F**). There were no sex differences or an interaction of irradiation × sex.

## Discussion

Brain tumors are the second most common type of pediatric cancer, and medulloblastoma/ primitive neuroectodermal tumors represent the most common malignant brain tumor in children. With a peak incidence in early childhood, medulloblastomas are potentially curable but generally require treatment with craniospinal radiation and chemotherapy, with significant late effects including risk for severe long-term neurocognitive complications [46]. Multiple studies have identified progressive decline in predicted intelligence quotient (IQ) after irradiation, with one study reporting an estimated decline in full-scale IQ after 4 years of 17.4 points [47]. Other studies have demonstrated similar findings with mean loss of 2.55 estimated full-scale IQ points per year related primarily to an inability to acquire new skills and information [48]. Cranial radiation is used in the treatment of other childhood brain tumors and leukemias, also with significant impact on neurocognitive outcomes. Contemporary approaches for the treatment of brain tumors in children and young adults have incorporated proton therapy in lieu of x-rays because of the precision afforded by protons in tumors requiring "boost" doses to tumor beds. A recent study of pediatric medulloblastoma patients revealed superior intellectual outcomes in those who received proton versus x-ray therapy likely associated with normal brain tissue sparing [49]. Global IQ, perceptual reasoning, and working memory outcomes were significantly better in the proton-treated group, and these patients had stable scores in these domains with the exception of processing speed. Proton radiation is emerging as the standard of therapy for medulloblastoma and other pediatric and young adult brain malignancies. Better understanding of the neurocognitive effects of proton therapy administered in conventional and FLASH dose rate is a high priority.

In our previous research with adult rats, proton doses of 11, 14, and 17 Gy delivered at 1 Gy/s led to a dose independent reduction in body weight compared with sham controls through the end of the experiment at 91 days post irradiation. In the current study, body weight reductions were not observed until 25 days post radiation treatment on P11, and the weight reductions were prominent after 8 Gy regardless of the delivery rate, whereas only the 5-Conv rats weighed less than controls at later time points. Sparing of body weight reductions were found for the 5-FLASH rats compared with control rats. Sparing of body weight reduction with a FLASH dose was found by others in a small sample of mice 9 days after proton irradiation at 16 Gy compared with the conventional dose rate [50]. The comparison of P11 irradiation with those in adult rats suggests that younger rats have a delayed and less severe decrease in body weight. This may be the result of the lower doses in the P11 rats or a factor of the rapid growth of the rats at this time. The mechanism underlying delayed weight gain in our studies was unclear.

In adult rats, 17 Gy whole brain irradiation with protons resulted in decreased total ambulation in an open field for up to 5 weeks [14]. In the current study with juvenile rats treated with a lower dose, there were no differences in total movement between irradiated groups and controls, although center ambulation was decreased in irradiated groups compared with controls, with relative sparing in the 5-FLASH group. In contrast, no differences in locomotion were

found in mice following cranial irradiation with electrons at conventional or FLASH rates [17]. Additionally, there was a small decrease in ASR and TSR for the 8-FLASH group compared with controls, but not for the other groups. When prepulses were added, no differences in performance were detected between irradiated groups and controls. In adult rats given conventional dose proton irradiation, no differences in ASR or TSR were observed, although there was increased responsiveness when prepulses were given [14]. Prior to learning in the water mazes, there were no differences on straight channel training trials between the irradiated groups and controls. These data suggest that although there was decreased activity observed in most exposure groups, there were no gross motor deficits on land or in the water for the irradiated rats compared with controls.

Egocentric learning and memory in the CWM is dependent, in part, upon the dopaminergic system of the striatum. For example, rats with 6-OHDA lesions of the neostriatum or nucleus accumbens, but not of the medial prefrontal cortex, have deficits in the CWM and impaired allocentric navigation in the MWM if the lesions are extensive [51–53]. However, dopamine depletion is not required for deficits in CWM learning. For example, deficits in the CWM were detected in rats that had no differences or increases in striatal dopamine levels compared with controls [29, 54–56]. Other factors such as altered release of dopamine or changes in the receptor system may also underlie deficits in the CWM [29, 57]. The P11 proton irradiated rats all had deficits in the CWM when tested in adulthood, regardless of dose or dose rate when tested in the maze the first time. However, the 5-FLASH group was similar to controls from test day 10–18 in latency to reach the platform and from 9–16 and day 18 for errors. Therefore, the 5-FLASH group was delayed in learning the CWM, but were proficient as controls in the second half of the test. Similarly, proton irradiated adult rats had CWM deficits [14]. In the adult rats, these deficits were accompanied by a number of dopaminergic changes, but the most consistent was a decrease in the dopamine transporter (DAT). Dopamine levels or release were not examined. In the current experiment, there were increases in DAT and DRD1 in all but the 5-FLASH group. These dopaminergic changes may have resulted in the learning deficits for all the proton exposed groups, with the exception of the 5-FLASH group. Some sex-specific changes in DRD1 and TH were also found, but these did not correspond with maze performance since there were no sex × exposure differences in the CWM or other behaviors. As others have suggested, interpretation of interactions not predicted may not be reliable [58]. Comparison of the direction of change in the adult and neonatal experiments suggests that opposite effects occurred in some cases, that is decreased dopaminergic changes in adults and increased changes in the neonates. Striatal neuronal development in rats is complete in the prenatal or early postnatal period by P3 [20–22]. However, while the dopaminergic neurons in rats are in place around the time of birth, the system does not reach adult levels until approximately P30 [59]. This difference in maturation may help explain the differences in the dopaminergic system outcomes in adult versus neonates. Moreover, these data demonstrate that regardless of the direction of the perturbation to the striatum, deficits in the CWM are detected. The rats exposed on P11 were also tested in a mirror image version of the CWM that requires the rats to extinguish the learned pathway of the original maze and learn a new path. Consistent with the learning profile at the end of testing in the initial CWM (configuration A), in the mirror version of the maze (configuration B) the 5-FLASH group performed similar to controls. However, the deficits in the other irradiated groups remained. These data suggest that the 5-FLASH group remembered the tasks requirements of the CWM, whereas the other irradiated groups still could not perform as well as controls. This is one of the first studies to use the mirror version of the CWM, a form of cognitive flexibility and reversal learning, and provides evidence that the impact on the striatum is long-lasting after proton irradiation.

The hippocampus is vulnerable to irradiation therapies and when possible is avoided in human treatment plans [1, 3]. Animal studies have also demonstrated that the hippocampus is vulnerable to irradiation [60]. Efficient learning in the MWM is dependent upon the hippocampus as well as other closely related brain regions. Adult rats treated with 11–17 Gy cranial irradiation with protons had deficits in MWM learning [14], however when 5 or 8 Gy protons were delivered at P11, no deficits were noted. The physical maturation of the hippocampus may be an important factor in the differences between adult and P11 vulnerabilities. For P11 rats, the hippocampus is still undergoing neurogenesis and differentiation that lasts until approximately P21 [20–22]. Since the hippocampus is not fully formed at this age and neuronal progenitor cells are active, there may be compensation for any irradiation induced apoptosis or cellular injury resulting in effects different from those in regions more fully formed such as the striatum. Some support for this comes from studies on hypoxia-ischemia brain injury in mice on P9, where neurogenesis, as identified by BRDU incorporation, is suppressed on P9 after hypoxia-ischemia, but increased on P21, completely opposite of what is observed in the uninjured brain [61, 62]. In agreement with the MWM data, there were also no differences in conditioned freezing or in NOR in the present study, tests that also rely upon the hippocampus. Furthermore, the NMDA-R1 levels were similar among all groups suggesting that at P11, the hippocampus is spared from the deleterious effects of proton radiation. In other studies, adult mice exposed to 10 Gy electron irradiation and tested for conditioned freezing in an extinction paradigm had deficits if the exposure rate was 1 Gy/s, but were protected if a FLASH rate was used [17]. Such data suggest that rats older than P30, when the brain of rats is more fully developed similar to peri-adolescence, may better model irradiation effects in children.

In an initial study on the FLASH effect with a 10 Gy whole brain electron irradiation in adult mice, there was a sparing of NOR when the dose rate was higher than 60 Gy/s, but not at rates 30 Gy/s and lower [19]. Subsequent studies in mice demonstrated that the FLASH sparing effect for electron irradiation was dose-dependent, such that at 10 or 12 Gy doses at 100 Gy/s there was sparing of NOR, but at 14 Gy no sparing was observed [16, 17]. Following proton irradiation on P11 here, regardless of dose or dose rate, there were no differences in NOR. Similarly, no differences in NOR were detected following adult rat proton irradiation at 1 Gy/s at doses between 11–17 Gy. There are several possibilities for the difference in NOR performance, such as the irradiation type, the species, age at irradiation, or the NOR protocol used in the study. The NOR test in radiation research and some of its limitations has been discussed [15]. NOR provides only a snapshot of incidental learning and does not provide information for other learning domains.

Not all dose-rates or doses produce a beneficial FLASH effect (reviewed by Montay-Gruel et al. [16]). After a 10 Gy dose of electron FLASH in female mice, sparing of NOR was observed for dose rates 60 Gy/s or greater but not for 30 Gy/s or less [19]. In this experiment, the 5-FLASH group had deficit sparing for central activity in the open field, CWM learning, RWM latency and errors on day 1, and DAT when compared with the 5-Conv group. In comparison, the 8-FLASH group had little to no sparing. Dose-dependent FLASH effects are found after electron exposure as well, with 10 to 12 Gy showing sparing of NOR but not 14 Gy [16, 17]. After a 10 Gy X-ray dose at 37 Gy/s, female mice had similar performance on NOR compared with controls [18]. Other domains of cognitive performance have not been examined thoroughly in adult rodents following FLASH radiotherapy, however some anxiety measures appear to be spared from deficits following FLASH doses of electrons [17]. Taken together, the 5-FLASH group was spared most of the deficits observed in the 5-Conv group, but there were little to no differences in the 8 Gy groups suggesting a dose-dependent effect for FLASH at this age.

Limitations of this experiment include delivering proton irradiation on only one age (P11), an age that is roughly equivalent in terms of brain ontogeny to a child about one year of age. Although this is younger than the age when most brain tumors occur in children, there are many instances when children under three years of age would benefit from radiotherapy to the whole CNS including the brain, and this treatment cannot be delivered due to the devastating neurocognitive toxicity that would result. Comparison of adult proton treatment and the P11 exposure shows that the severity of deficits are related to brain maturity. Therefore, examination of other ages and stages of brain development would be informative. Another limitation was the use of only one FLASH dose rate, precluding the ability to identify a threshold or gradient in the protective effect of FLASH with respect to neurocognitive toxicity. Thus, to fully understand the FLASH effect other rates should be assessed for comparison, however for our study we felt it was important to first establish that the model results in learning and memory deficits before modifying additional variables such as dose rate. Given that 5 Gy and 8 Gy induced deficits in learning and memory, these doses given at different rates could provide useful information. Determining the effect of fractionating the treatment into smaller doses to better represent the radiotherapy used for patient treatments should be investigated, but this was beyond the range of the present experiment. Others have shown that fractionation with FLASH produces similar sparing of NOR as a single dose at FLASH rates [16]. Finally, the rats in this study did not have tumors. A rat model that has tumors in combination with FLASH vs. conventional rates and examines the ensuing cognitive deficits is needed.

## Conclusions

In conclusion, neurocognitive toxicity after cranial irradiation with protons was reduced when the 5 Gy dose was delivered at a rate of 100 Gy/s. In agreement with previous studies, there appears to be a limit to the FLASH sparing effectiveness. For example, FLASH had sparing effects at a dose of 10 Gy but not 14 Gy after electron irradiation [16, 17], whereas in the current study, FLASH had sparing effects at 5 Gy but not 8 Gy. The dopaminergic system appears vulnerable to the effects of proton irradiation in young and adult rats. While concerns have been raised in regard to the proton radiation sensitivity of the hippocampus, the present data, at least for P11 exposure, suggest greater vulnerability of the striatum than the hippocampus. This may reflect hippocampal compensation since at P11 neurogenesis is ongoing, hence, radiation damaged cells may get replaced after P11 leaving little residual effect. In summary, FLASH proton irradiation appears to provide some degree of neuroprotection, however, additional investigations to identify parameters (e.g. dose, dose-rate, irradiation age, specific neurocognitive outcomes) for optimal benefit and minimal cognitive side-effects are needed.

## Supporting information

**S1 Data.**
(ZIP)

## Acknowledgments

We thank Erin Tepe, Audra Stueve, Jacob Feldman, Sydney Bevelheimer, Aliyah Lingo, Shalyn Brown, Kaitlyn Lohman, Caitlin Lachut, and Glen McClain for assisting in testing of rats used in this research.

## Author Contributions

**Conceptualization:** Michael T. Williams, Ralph E. Vatner, John P. Perentesis, Charles V. Vorhees.

**Data curation:** Chiho Sugimoto, Adam L. Fritz.

**Formal analysis:** Michael T. Williams.

**Funding acquisition:** Michael T. Williams, John P. Perentesis, Charles V. Vorhees.

**Investigation:** Chiho Sugimoto, Samantha L. Regan, Emily M. Pitzer, Adam L. Fritz.

**Project administration:** Michael T. Williams, Charles V. Vorhees.

**Resources:** Mathieu Sertorio, Anthony E. Mascia.

**Supervision:** Michael T. Williams, Charles V. Vorhees.

**Visualization:** Michael T. Williams, Charles V. Vorhees.

**Writing – original draft:** Michael T. Williams, Charles V. Vorhees.

**Writing – review & editing:** Michael T. Williams, Chiho Sugimoto, Samantha L. Regan, Emily M. Pitzer, Adam L. Fritz, Mathieu Sertorio, Anthony E. Mascia, Ralph E. Vatner, John P. Perentesis, Charles V. Vorhees.

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
