## [Decision Letter · Decision Letter 0]

7 Jun 2022

PONE-D-22-04666

Cognitive and behavioral effects of whole brain conventional or high dose rate (FLASH) proton irradiation in a neonatal Sprague Dawley rat model

PLOS ONE

Dear Dr. Williams,

Thank you for submitting your manuscript to PLOS ONE. After careful consideration, we feel that it has merit but does not fully meet PLOS ONE’s publication criteria as it currently stands. Therefore, we invite you to submit a revised version of the manuscript that addresses the points raised during the review process.

Please address the following issues raised by the reviewers

- Reviewer 1: Provide additional explanation for the limited protective effect of FLASH , which incorporates findings from other groups. The results of the ELISA and Western blot analysis should also be mentioned in the discussion.

- Reviewer 2: Please provide a rationale for the order of the behavioural tasks particularly in regards to the ASR and TSR being close to the start of the testing battery

- Reviewer 3:

1) While the days on which the behavioral tests are described in each section, it would be helpful to have a timeline figure that shows the organization of these days over time. This gives the reader a better idea of the differences in time needed for each behavioral test.

2) The authors should comment on the potential for numerous behavioral tests to interact with or cause changes in future behavioral performance. For example, they noted that the behavioral tests were run from assumed least stressful to most stressful, but it is possible that learning one test could alter learning of subsequent behavioral tests. Given that only one test was repeated and the results were different from the first assessment (5 Gy FLASH), would differences in performances of the other groups potentially be reversed if the rats were re-tested?

3) There were limited effects on the neuronal markers chosen (DRD1, DRD2, DAT, TH, NMDA-NR1). The authors should comment on the lack of measurements typical of other FLASH studies, such as neuroinflammation and neurogenesis and what role these processes might play in the differences between these groups.

We look forward to receiving your revised manuscript.

Kind regards,

Joohyung Lee, PhD

Academic Editor

PLOS ONE

**Journal requirements:**

“This research was supported by a Research Agreement from Varian, a Siemens Healthineers company. Additional funding support included research support from the TQL Foundation, Cincinnati, OH and Fund the Cure Next Door Foundation.”

This research was supported by a Research Agreement from Varian, a Siemens Healthineers company. Additional funding support included research support from the TQL Foundation, Cincinnati, OH and Fund the Cure Next Door Foundation.”

“This research was supported by a Research Agreement from Varian, a Siemens Healthineers company. Additional funding support included research support from the TQL Foundation, Cincinnati, OH and Fund the Cure Next Door Foundation.”

“These experiments were funded by Varian, a Siemens Healthineers company that granted the authors intellectual freedom to publish the data.”

Reviewers' comments:

Reviewer's Responses to Questions

**Comments to the Author**

1. Is the manuscript technically sound, and do the data support the conclusions?

Reviewer #1: Yes

Reviewer #2: Yes

Reviewer #3: Yes

2. Has the statistical analysis been performed appropriately and rigorously? 

Reviewer #1: Yes

Reviewer #2: Yes

Reviewer #3: Yes

3. Have the authors made all data underlying the findings in their manuscript fully available?

Reviewer #1: Yes

Reviewer #2: Yes

Reviewer #3: Yes

4. Is the manuscript presented in an intelligible fashion and written in standard English?

Reviewer #1: Yes

Reviewer #2: Yes

Reviewer #3: Yes

5. Review Comments to the Author

Reviewer #1: This is a well designed, careful neurotoxicity investigation of whole brain irradiation using 11 days old rats comparing the neurognitive and behavioral effects and concentration of neurochemical markers after conventional and FLASH proton single dose delivery at two dose levels (5 and 8 Gy).

The authors revealed important findings on the vulnerability of the young rat brain, identified underlying pathomechanism of cognitive and behavioural changes, but found only modes protective effect of the FLASH irradiation, and only at the 5 Gy dose level. They pointed out in the manuscrip correcty the limitations of the experiment, but these limitations are not explaine completely the lack of more pronounced protective effect of the FLASH. The reasons for that and the more evident results of other groups should be discussed in more details. As well as the results of the ELISA and Western blot analysis should be mentioned in the discussion.

If some senteces will be added on these points to the discussion I fully support the publication of this valuable manuscript.

Reviewer #2: Fantastic study and in-depth analysis of many crucial cognitive domains as a part of your behavioural suite, well done. The statistical analysis was also great to show the level of detail, and made all relevant comparisons for not only your hypotheses, but the validity of the measures. One thing that came to mind was the order of the behavioural tasks - I was intrigued to see ASR and TSR so close to the start of the testing battery. My belief was that exposure to the acoustic startle stimuli in the ASR (not sure about TSR) changes the underlying neurobiology of the dopamine system, so that it is different post-exposure to if the animals are naiive to this task. In this way, ASR is typically always measured as the last task? Overall, great quality paper for submission number 1.

Reviewer #3: The current manuscript details an experiment comparing conventionally delivered doses of protons (5 and 8Gy) and the same doses delivered using FLASH dose rates. Overall, the manuscript is well-written and provides a number of behavioral assays covering multiple neurobehavioral domains. I have a few minor comments.

1) While the days on which the behavioral tests are described in each section, it would be helpful to have a timeline figure that shows the organization of these days over time. This gives the reader a better idea of the differences in time needed for each behavioral test.

2) The authors should comment on the potential for numerous behavioral tests to interact with or cause changes in future behavioral performance. For example, they noted that the behavioral tests were run from assumed least stressful to most stressful, but it is possible that learning one test could alter learning of subsequent behavioral tests. Given that only one test was repeated and the results were different from the first assessment (5 Gy FLASH), would differences in performances of the other groups potentially be reversed if the rats were re-tested?

3) There were limited effects on the neuronal markers chosen (DRD1, DRD2, DAT, TH, NMDA-NR1). The authors should comment on the lack of measurements typical of other FLASH studies, such as neuroinflammation and neurogenesis and what role these processes might play in the differences between these groups.

6. PLOS authors have the option to publish the peer review history of their article (what does this mean?). If published, this will include your full peer review and any attached files.

Reviewer #1: No

Reviewer #2: No

Reviewer #3: No

---

## [Author Response · Author response to Decision Letter 0]

27 Jul 2022

Reviewer 1: Provide additional explanation for the limited protective effect of FLASH, which incorporates findings from other groups. The results of the ELISA and Western blot analysis should also be mentioned in the discussion.

Reply. A new paragraph has been added to the Discussion on the protective effects. We argue that the 5-FLASH group produced sparing when there were deficits in the 5-Conv group. We have also incorporated the results of the western and ELISAs.

- Reviewer 2: Please provide a rationale for the order of the behavioural tasks particularly in regards to the ASR and TSR being close to the start of the testing battery

Reply: The following has been added: “A behavioral battery of tests was used to reduce the number of rats used, provide a better characterization of effects for this initial experiment (41), and to be more translatable to human studies that also employ test batteries of different cognitive domains. While test order may have some effect on subsequent behaviors (42), having a day between different tests alleviated many of these issues (43). We also minimized these effects by performing tasks from the least stressful to most stressful (with the exception of the CWM mirror version). While fear potentiated startle can cause changes in dopamine (44), the startle response itself is not known to produce long-term dopamine changes (45).” 

- Reviewer 3:

1) While the days on which the behavioral tests are described in each section, it would be helpful to have a timeline figure that shows the organization of these days over time. This gives the reader a better idea of the differences in time needed for each behavioral test.

Reply: A timeline is now provided.

2) The authors should comment on the potential for numerous behavioral tests to interact with or cause changes in future behavioral performance. For example, they noted that the behavioral tests were run from assumed least stressful to most stressful, but it is possible that learning one test could alter learning of subsequent behavioral tests. Given that only one test was repeated and the results were different from the first assessment (5 Gy FLASH), would differences in performances of the other groups potentially be reversed if the rats were re-tested?

Reply: There were 2 different versions of the Cincinnati water maze (CWM) used in this experiment. For the first version, the 5-FLASH group was delayed in learning, as demonstrated by the interaction of exposure x day, but performed equally to the controls by the end of the test. We now make this point better in the Discussion. The mirror version of the CWM while similar requires a completely new set of turns and the 5-FLASH group performed similarly to controls. We have now made this clear that there is consistency in the effects that were observed for the 5-FLASH group. 

3) There were limited effects on the neuronal markers chosen (DRD1, DRD2, DAT, TH, NMDA-NR1). The authors should comment on the lack of measurements typical of other FLASH studies, such as neuroinflammation and neurogenesis and what role these processes might play in the differences between these groups.

Reply: We have added the following: A number of studies examining the effect of FLASH in the brain of adult mice have focused on markers of overt neurotoxicity such as glial fibrillary acidic protein staining or bromodeoxyuridine uptake (16-18). However, in this study the approach was to examine the neurotransmitter system related to the functional changes observed in these rats. Therefore the focus was on the dopamine system because of the changes in the CWM and the NMDA receptor subunit 1 (NMDA-NR1) in the hippocampus to confirm the MWM results. 

Reply: style changed as required.

“This research was supported by a Research Agreement from Varian, a Siemens Healthineers company. Additional funding support included research support from the TQL Foundation, Cincinnati, OH and Fund the Cure Next Door Foundation.”

Reply: Funding-related text was removed from the manuscript. The Funding Statement should reads as follows:

This research was supported by a Research Agreement from Varian, a Siemens Healthineers company. Additional funding support included research support from the TQL Foundation, Cincinnati, OH and Fund the Cure Next Door Foundation.

“This research was supported by a Research Agreement from Varian, a Siemens Healthineers company. Additional funding support included research support from the TQL Foundation, Cincinnati, OH and Fund the Cure Next Door Foundation.”

Reply: The financial disclosure should read as follows:

This research was supported by a Research Agreement from Varian, a Siemens Healthineers company. Additional funding support included research support from the TQL Foundation, Cincinnati, OH and Fund the Cure Next Door Foundation. The funders had no role in study design, data collection and analysis, decision to publish, or preparation of the manuscript. Feedback was provided by Varian for the initial manuscript submission. 

“These experiments were funded by Varian, a Siemens Healthineers company that granted the authors intellectual freedom to publish the data.”

Reply: The Competing Interests statement should read as follows:

These experiments were funded by Varian, a Siemens Healthineers company that granted the authors intellectual freedom to publish the data. This does not alter our adherence to PLOS ONE policies on sharing data and materials.

Reply: There are no in text citations to supporting information.

---

## [Decision Letter · Decision Letter 1]

22 Aug 2022

Cognitive and behavioral effects of whole brain conventional or high dose rate (FLASH) proton irradiation in a neonatal Sprague Dawley rat model

PONE-D-22-04666R1

Dear Dr. Williams,

We’re pleased to inform you that your manuscript has been judged scientifically suitable for publication and will be formally accepted for publication once it meets all outstanding technical requirements.

Kind regards,

Joohyung Lee, PhD

Academic Editor

PLOS ONE

Additional Editor Comments (optional):

Reviewers' comments:

Reviewer's Responses to Questions

**Comments to the Author**

1. If the authors have adequately addressed your comments raised in a previous round of review and you feel that this manuscript is now acceptable for publication, you may indicate that here to bypass the “Comments to the Author” section, enter your conflict of interest statement in the “Confidential to Editor” section, and submit your "Accept" recommendation.

Reviewer #2: All comments have been addressed

Reviewer #3: All comments have been addressed

2. Is the manuscript technically sound, and do the data support the conclusions?

Reviewer #2: Yes

Reviewer #3: Yes

3. Has the statistical analysis been performed appropriately and rigorously? 

Reviewer #2: Yes

Reviewer #3: Yes

4. Have the authors made all data underlying the findings in their manuscript fully available?

Reviewer #2: Yes

Reviewer #3: Yes

5. Is the manuscript presented in an intelligible fashion and written in standard English?

Reviewer #2: Yes

Reviewer #3: Yes

6. Review Comments to the Author

Reviewer #2: All comments from all reviewers have been addressed well, strengthening the overall quality of the paper. Well done.

Reviewer #3: (No Response)

7. PLOS authors have the option to publish the peer review history of their article (what does this mean?). If published, this will include your full peer review and any attached files.

Reviewer #2: No

Reviewer #3: No

---

## [Editor Report · Acceptance letter]

9 Sep 2022

PONE-D-22-04666R1 

Cognitive and behavioral effects of whole brain conventional or high dose rate (FLASH) proton irradiation in a neonatal Sprague Dawley rat model 

Dear Dr. Williams:

I'm pleased to inform you that your manuscript has been deemed suitable for publication in PLOS ONE. Congratulations! Your manuscript is now with our production department. 

Kind regards, 

on behalf of

Dr Joohyung Lee 

Academic Editor

PLOS ONE